# A spatially-resolved transcriptional atlas of the murine dorsal pons at single-cell resolution

Stefano Nardone[1,2], Roberto De Luca [3,12], Antonino Zito[4,10,11,12], Nataliya Klymko[5,12], Dimitris Nicoloutsopoulos[6], Oren Amsalem[1], Cory Brannigan[7], Jon M. Resch [8,9], Christopher L. Jacobs[1,2], Deepti Pant[1], Molly Veregge[1], Harini Srinivasan[1,2], Ryan M. Grippo[1], Zongfang Yang[1], Mark L. Zeidel[5], Mark L. Andermann [1], Kenneth D. Harris [6], Linus T. Tsai [1,2], Elda Arrigoni[3], Anne M. J. Verstegen [5] ✉, Clifford B. Saper [3] ✉ & Bradford B. Lowell [1] ✉

The "dorsal pons", or "dorsal pontine tegmentum" (dPnTg), is part of the brainstem. It is a complex, densely packed region whose nuclei are involved in regulating many vital functions. Notable among them are the parabrachial nucleus, the Kölliker Fuse, the Barrington nucleus, the locus coeruleus, and the dorsal, laterodorsal, and ventral tegmental nuclei. In this study, we applied single-nucleus RNA-seq (snRNA-seq) to resolve neuronal subtypes based on their unique transcriptional profiles and then used multiplexed error robust fluorescence in situ hybridization (MERFISH) to map them spatially. We sampled ~1 million cells across the dPnTg and defined the spatial distribution of over 120 neuronal subtypes. Our analysis identified an unpredicted high transcriptional diversity in this region and pinpointed the unique marker genes of many neuronal subtypes. We also demonstrated that many neuronal subtypes are transcriptionally similar between humans and mice, enhancing this study's translational value. Finally, we developed a freely accessible, GPU and CPU-powered dashboard (http://harvard.heavy.ai:6273/) that combines interactive visual analytics and hardware-accelerated SQL into a data science framework to allow the scientific community to query and gain insights into the data.

The pons consists of two main divisions: the "pontine tegmentum", which represents its dorsal part, and the "basis pontis", which is its ventral part. This study focuses on the dorsal portion of the pontine tegmentum (dPnTg). The dPnTg plays a pivotal role in the functioning of the autonomic nervous system, but it also represents a strategic hub for integrating many vital processes. It harbors many anatomically defined subnuclei (Table 1) that perform a wide range of functions, including the PB and pre-LC, which have been implicated in receiving ascending visceral sensory and pain inputs from the spinal cord and medulla, and integrating them with forebrain cognitive, arousal, and emotional inputs to direct behavior, autonomic, and endocrine functions. In addition, specific neuronal populations residing in this area have been reported to be involved in respiration[1,2], arousal[3–5] sleep-wake regulation[6], pain[7,8], reward processing and reinforcement[9–11], movement[12,13], memory formation[14], feeding[15,16], micturition[17,18], aversive behaviors[19], thermoregulation[20], cardiovascular regulation[21], itch[22], and other behaviors. To facilitate future mechanistic investigations of how this brain region controls these processes, it is of great

**Table 1 | Abbreviations of dPnTg brain regions**

| abbreviation | brain nucleus | abbreviation | brain nucleus |
|---|---|---|---|
| Bar | Barrington's nucleus | LPBI | LPB, internal part |
| CGA | Central gray, alpha part | LPBS | LPB, superior part |
| CGB | Central gray, beta part | LPBV | LPB, ventral part |
| CGPn | Central gray of the pons | Me5 or MTN | Mesencephalic trigeminal nucleus |
| DR | Dorsal raphe nucleus | MnR | Median raphe nucleus |
| DTgC | Dorsal tegmental nucleus, central part | MPB | Medial parabrachial nucleus |
| DTgP | Dorsal tegmental nucleus, pericentral part | MPBE | MPB, external part |
| KF | Kölliker-Fuse nucleus | O or NI | Nucleus O or nucleus incertus |
| LC | Locus coeruleus | PB | Parabrachial nucleus |
| LDTg | Laterodorsal tegmental nucleus | PBW | Parabrachial nucleus, waist part |
| LDTgV | Laterodorsal tegmental nucleus, ventral | PDTg | Posterodorsal tegmental nucleus |
| LPB | Lateral parabrachial nucleus | PPTg | Pedunculopontine tegmental nucleus |
| LPBC | LPB, central part | pre-LC | Pre locus coeruleus |
| LPBCr | LPB, crescent part | Sph | Sphenoid nucleus |
| LPBD | LPB, dorsal part | SPTg | Subpedencular tegmental nucleus |
| LPBE | LPB, external part | VTg | Ventral tegmental nucleus |

interest to catalog, at a transcriptional level, all the neuron subtypes that populate this area.

Here, we applied single-nucleus and spatial transcriptomics to unravel the neuronal complexity of the dPnTg[23,24]. To accomplish this, we first performed snRNA-seq on cells from this region. The purpose of this first step was two-fold: to identify highly informative marker genes specifying each neuronal subtype, which we would later use for spatial localization, and to obtain a complete transcriptomic inventory of genes expressed by the different neuronal subtypes. Then, MERFISH was performed using 315 informative genes to spatially locate each neuronal subtype within the dPnTg.

## Results

### Single-nucleus transcriptional profiling identifies distinct cell types in the dPnTg

To profile the single-nuclei whole transcriptome of the dPnTg, we employed two snRNA-seq approaches: DroNc-seq[23] and 10X (Fig. 1a). DroNc-seq data were generated by this study using tissue biopsies restricted to dPnTg, whereas the 10X data were retrieved from the Allen Brain Atlas (ABA) effort that used tissue biopsies representing the entire pons[25,26]. To dissect the dPnTg with high precision, in the DroNc-seq dataset we marked the PB and Bar, two brain nuclei that help define its extent, and used their fluorescent signal to guide the dissection (Fig. 1b; methods), whereas in the 10X dataset of the pons, we used the anatomical annotation available for each nucleus, imputed from MERFISH data, to select only nuclei belonging to the dPnTg. After preprocessing and quality control steps, a merged dataset of 222,592 nuclei x 34,457 genes was analyzed using a pipeline that includes Seurat v.3.2.3 and Harmony v.1.1 packages[27–30] (Supplementary Fig. 1a, b; methods). Our analysis identified 63 clusters comprising 12 major cell types (Fig. 1c, d; Supplementary Fig. 2a). Each cell type was characterized by uniquely expressed genes (i.e., markers), of which many have been previously reported in the literature (Fig. 1e; Supplementary Fig. 2b; Supplementary Data 1). Neurons encompassed 32 clusters, accounting for ~40% of all nuclei. The glial/non-neuronal cells encompassed 31 clusters, accounting for the remaining 60% of all nuclei (Fig. 1c, d; Supplementary Fig. 2a). We identified 11 major glial/non-neuronal cell types: oligodendrocytes, astrocytes, oligodendrocyte precursor cells (OPCs), immature oligodendrocytes, perivascular macrophages (PVMs), microglia, vascular smooth muscle cells (VSMCs), pericytes, vascular and leptomeningeal cells (VLMCs type I

and II), choroid plexus epithelial cells (CPE) and ependymocytes (Fig. 1c–e; Supplementary Fig. 2a, b; Supplementary Data 1).

To disentangle the neuronal diversity of the dPnTg, we first selected all neurons, excluding neurons outside our region of interest (ROI) and glial/non-neuronal cells, and then we categorized them into two main groups for re-clustering (methods). The first group, called "excitatory neurons", included 47,756 nuclei divided into 71 clusters. They expressed *Slc17a6*, *Slc17a7*, or *Slc17a8* (glutamatergic neurons), and in some cases, they expressed *Th*/*Slc18a2* (noradrenergic neurons), *Tph2*/*Slc6a4* (serotoninergic neurons), *Chat*/*Slc5a7* (cholinergic neurons) or *Slc17a6*/*Slc32a1* ("hybrid neurons")[24] (Fig. 1f, h). The second group, "inhibitory neurons", included 30,771 nuclei divided into 57 clusters. All neurons in this group expressed *Slc32a1* (GABAergic neurons), and at the same time, some also expressed *Slc6a5* (glycinergic neurons) (Fig. 1g, i). Each cluster was defined by the expression of one or a combination of marker genes (Fig. 1g–i; Supplementary Data 2, 3). Albeit to a different extent, every covariate contributed to each neuronal cluster (% of cells), confirming the mitigation of the batch effects (Supplementary Fig. 2c–i). Our analysis pinpointed many neuronal types and confirmed several already documented in the literature, identifying even rare populations accounting for <1% of cells in the dataset.

### MERFISH allows the identification and localization of distinct cell types in the dPnTg

We employed MERFISH to spatially resolve the transcriptional neuronal organization of the dPnTg (Fig. 2a). Specifically, we investigated the spatial patterns of 315 genes that include (1) marker genes from differential expression (DE) analysis of the snRNAseq dataset, (2) canonical glial, non-neuronal, and neuronal markers; and (3) transcription factors, neuropeptides, and receptors (Supplementary Data 4; methods). We profiled ~5.5 million cells across 46 coronal sections from 7 mice spanning, at intervals of 80–90 μm, a brain region corresponding to −4.7 to −5.8 bregma level in the Franklin-Paxinos atlas[31]. For each MERFISH section, we manually defined the boundaries of the ROI, i.e., dPnTg. The dorsal boundary at rostral levels was defined by the inferior colliculus and more caudally by the dorsal surface of the pons; the ventral boundary was the dorsal part of the motor trigeminal nucleus (Mo5). Then, we used the boundaries' pixel cartesian coordinates to subset each gene counts matrix to include only cells (polygons) and transcripts (spots) inside the ROI. After removing low-

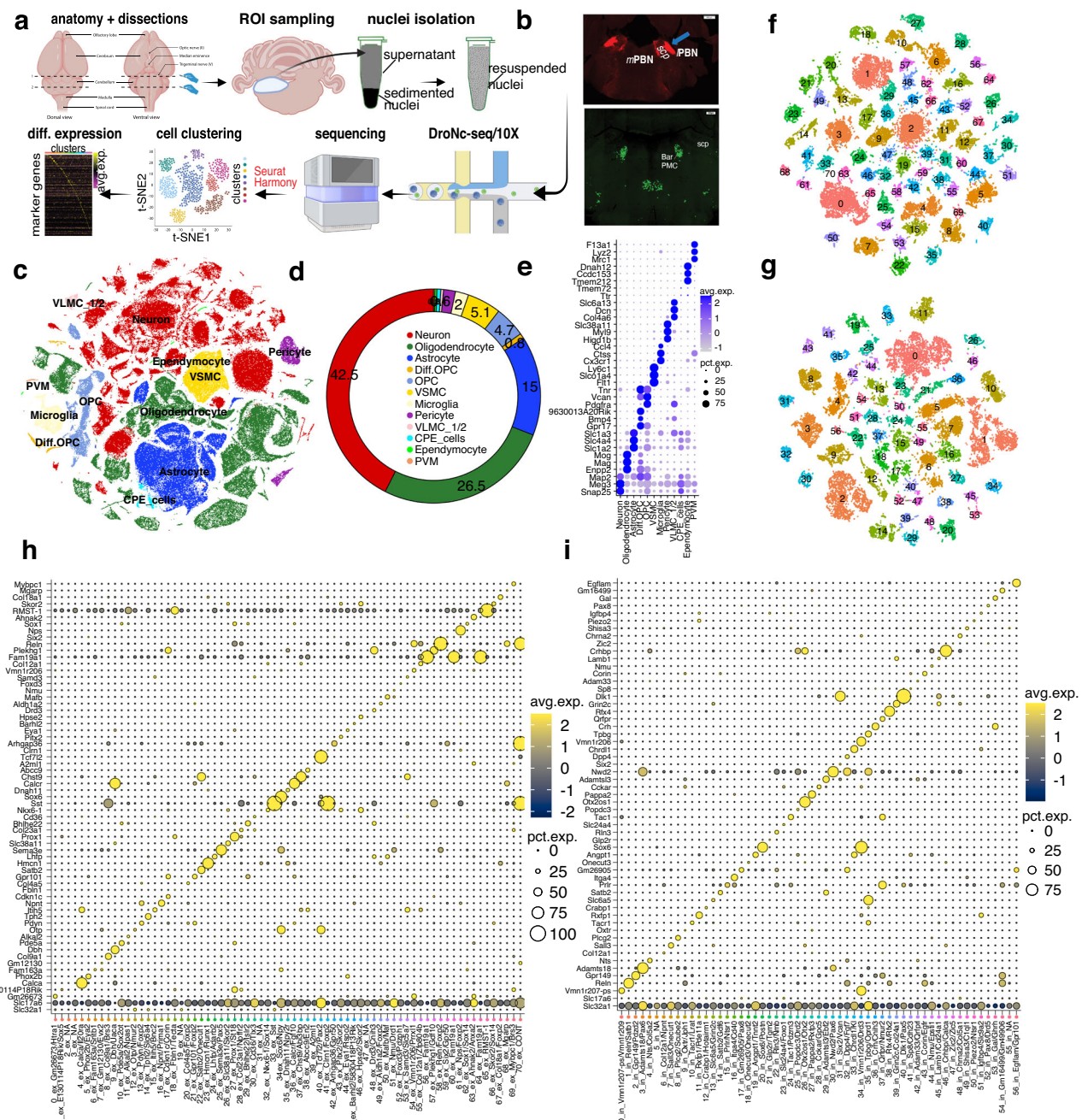

**Fig. 1 | snRNA-seq transcriptional profile of the dPnTg. a** Experimental workflow summarized in five main steps: brain dissections, nuclei isolation, snRNA-seq, sequencing, and bioinformatic analyses. **b** Image illustrating the two dissection strategies relying on the visualization of PB (top) and Bar (bottom). (scale bar: 500 μm). **c** t-SNE plot of 222,592 nuclei color-coded according to the legend in (**d**). **d** Donut plot representing the fraction (%) of each cell type identified. **e** Dot plot of 35 cell marker genes that univocally identify each cell type. 3 marker genes were plotted for all cell types except for CPE cells, where only the top 2 were used. **f**, **g** t-SNE plots showing 47,756 nuclei from the "excitatory" group (**f**) and 30,771 nuclei from the "inhibitory" group (**g**) color-coded by cell cluster. The top marker genes that specify the identity of each "excitatory" or "inhibitory" cluster are in (**h**)

and (**i**), respectively. **h**, **i** Dot plots illustrating the expression level of the top marker gene for the "excitatory" (**h**) and "inhibitory" (**i**) neuronal groups. All differentially expressed genes in the dot plot have an average log fold-change >0.25 and an adjusted *p*-value <0.01. Test used: *Wilcoxon Rank Sum two-sided Bonferroni-corrected Test*. lPBN/mPBN parabrachial nucleus lateral/medial divisions; Bar, Barrington's nucleus; scp, superior cerebellar peduncle; t-SNE, t-distributed Stochastic Neighbor Embedding; OPC, oligodendrocyte progenitor cell; PVM, perivascular macrophages; VSMC, vascular smooth muscle cells; CPE cells, choroid plexus epithelial cells; VLMC1/2, vascular and leptomeningeal cell type 1/2; Diff.OPC, immature oligodendrocytes; NA, no marker detected; CONT, glia contamination. Figure 1a was generated using BioRender.

quality and external-to-ROI cells, 685,289 cells were retained for downstream analyses (methods). Throughout the manuscript, all mention of rostral to caudal bregma levels refers to sections approximated to the best matched in the Franklin-Paxinos atlas[31]. We also used the nomenclature from that atlas to identify nuclei and brain areas.

Our analysis of all cells from the ROI identified 44 clusters grouped into 9 transcriptionally distinct cell types. Neurons encompassed 24 clusters, accounting for 50% of all cells in the dataset (Fig. 2b, c; Supplementary Fig. 7a; Supplementary Data 5). Each cell type was characterized by uniquely expressed genes (Fig. 2d; Supplementary Fig. 7b). Afterward, we selected only the

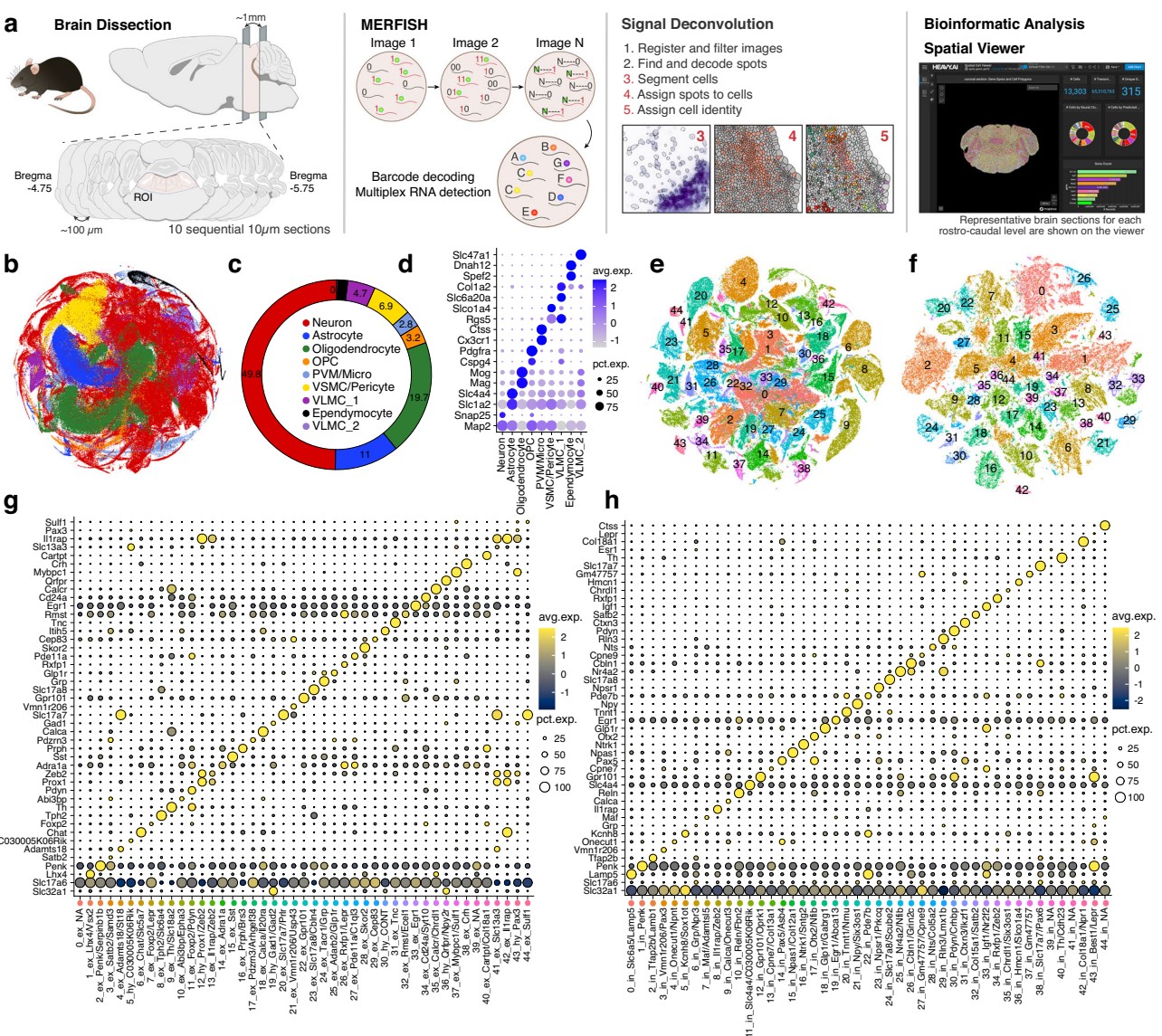

**Fig. 2 | MERFISH transcriptional profile of the dPnTg. a** Experimental workflow summarized in five main steps: brain dissection, MERFISH assay, signal deconvolution, bioinformatic analyses, and data visualization. In total, 7 animals were used, of which 4 represent a complete series of 10 serial coronal sections. **b** t-SNE plot of 685,289 cells color-coded according to the legend in (**c**). **c** Donut plot depicting the fraction (%) of each cell type identified. **d** Dot plot of 17 cell markers (y-axis) that univocally identify each cell type (x-axis). For each cell type, 2 markers were plotted, except for VLMC types I and II, where 1 marker was used. **e**, **f** t-SNE of 231,103 cells from the "excitatory" group (**e**) and 110,332 cells from the "inhibitory" group (**f**) color-coded by cell cluster. The top 2 marker genes specify the identity of each

cluster as per (**g**) and (**h**), respectively. **g**, **h** Dot plot of the expression level of the top marker gene for the "excitatory" (**g**) and "inhibitory" (**h**) neuronal clusters. All differentially expressed genes in the dot plot have an average log fold-change >0.25 and an adjusted p-value < 0.01. Test used: *Wilcoxon Rank Sum two-sided Bonferroni-corrected Test*. t-SNE, t-distributed Stochastic Neighbor Embedding; OPC, oligodendrocyte progenitor cell; PVM, perivascular macrophages; VSMC, vascular smooth muscle cells; CPE cells, choroid plexus epithelial cells; VLMC1/2, vascular and leptomeningeal cell type 1/2; Diff.OPC, immature oligodendrocytes; NA, no marker detected; CONT, glia contamination.

neurons, discarded the glial/non-neuronal clusters, and, as before, divided them into two main groups for re-clustering. The first group, called "excitatory neurons", included 231,103 cells divided into 45 clusters (Fig. 2e). The second group, called "inhibitory neurons", included 110,332 cells divided into 45 clusters (Fig. 2f). Each cluster was defined by the expression of one or a combination of marker genes (Fig. 2g, h; Supplementary Data 6, 7). Cells from different MERFISH slides belonging approximately to the same rostrocaudal level contributed equally to the same neuronal clusters (% of cells), confirming the reproducibility between independent series of sections (Supplementary Fig. 7c, d). Both genders were equally represented among the clusters (Supplementary Fig. 7e). Supplementary Data 8 ("excitatory neurons") and 9 ("inhibitory neurons")

comprehensively list the neuronal MERFISH clusters, their marker genes, and spatial location. Finally, to provide transcriptional resolution on a spatial scale that is of specific interest to investigators and achieve better cluster granularity, we re-clustered the MERFISH-profiled neurons according to four anatomically defined subregions that include the following brain nuclei: (1) KF; (2) LPB and MPB; (3) MTN, pre-LC, LC, and Bar; and (4) LDTgV, LDTg, VTg, DTgC, DTgP, PDTg, CGA, CGB, Sph, O, and CGPn (Table 1). In this study, we excluded from downstream analyses brain nuclei that were only partially represented within the ROI in our sections (e.g., DR, PPTg, SPTg). To avoid ambiguity in the cluster nomenclature, we prepended a prefix to each cluster ID for each subregion (as identified above): "at1_", "at2_", "at3_", and "at4_", respectively.

## MERFISH-resolved atlas of the KF

The KF, along with the LPB and MPB, is one of the three subdivisions of the parabrachial complex and is predominantly located in sections just rostral to the LPB and MPB[32,33]. To build a transcriptional atlas of the KF, first, we bilaterally traced its boundaries on MERFISH coronal sections spanning from −4.8 to −4.9 bregma level, and then, we used their pixel cartesian coordinates to subset each gene counts matrix to include only cells and transcripts inside the defined boundaries. A final dataset of 4554 neurons was analyzed using our bioinformatic pipeline (methods). This analysis pinpointed 19 clusters characterized by unique marker genes, which we classified into five groups based on shared gene expression profiles (Fig. 3a–c, f). Briefly, group 1 includes *Tfap2b*+ clusters at1_0, at1_1, at1_6, at1_8, at1_14, and at1_17; group 2 includes *Calca*+/*Onecut3*+ clusters at1_10 and at1_11; group 3, the only GABAergic/ glycinergic group, includes *Pax2*+ clusters at1_4 and at1_13; group 4 includes clusters at1_7 (*Nos1*+/*Lhx9*+) and at1_15 (*Nps*+/*Qrfpr*+)[34], both located outside the KF along the margin of the nucleus of the lateral lemniscus (NLL); lastly, the miscellaneous group includes clusters at1_2, at1_3, at1_5, at1_9, at1_12, at1_16, and at1_18, of which cluster at1_3 is located outside the KF (Fig. 3b–d; Supplementary Data 10). Next, to visualize neuronal clusters in space, we plotted the cartesian pixel coordinates of each cell as Voronoi plots and computed the cell frequency (cluster trajectory) across three bregma levels, from −4.8 to −4.9 (Fig. 3b, e). Interestingly, the four KF groups displayed distinct spatial distributions.

Then, we focused on *Calca*+ neurons, a well-known population of the LPBE[19,35,36], and hypothesized that *Calca*+ clusters at1_10 and at1_11 in the KF could be a more rostral continuation of that cell group. To test for this assumption, we assessed the transcriptional similarity by performing a Pearson's *r* correlation among the average expression of 315 genes across all neurons of KF clusters at1_10, at1_11, and at1_6 (negative control, *Calca*−) and the PB cluster at2_2. Strikingly, the KF cluster at1_10 exhibited the highest correlation score ($r = 84.8\%$) with PB cluster at2_2 compared to KF clusters at1_11 ($r = 59.6\%$) and at1_6 ($r = 33.3\%$) (Fig. 3g). While cluster at1_11 is scattered, cluster at1_10 is focally concentrated in the ventral part of the KF and could represent a rostral continuation of the main *Calca*+ LPBE population (Fig. 3i, j). To discover genetic markers that allow selective access to these neuronal subtypes, we performed a DE analysis between the PB cluster at2_2, all PB clusters except at2_2, and KF clusters at1_6, at1_10, and at1_11. *Calca* was expressed in KF clusters at1_10, at1_11, and PB cluster at2_2. *Onecut3* emerged as the most selective marker for KF *Calca*+ clusters at1_10 and at1_11 versus LPBE cluster at2_2. In addition, the genes *Ebf2* and *Chst9* selectively marked the KF cluster at1_11 (Fig. 3h; Supplementary Data 11). Anatomically, the KF clusters at1_10 and at1_11 mingle along their caudal edge with the most rostral neurons of the LPBE cluster at2_2. However, in the MERFISH assay, the KF *Calca*+ neurons express lower levels of *Calca* transcript and are smaller. In addition, using mice expressing Cre recombinase under the Calca promoter, *Calca* neurons in the PB complex have been found to project to the forebrain but also to the ventrolateral medulla[37]. Because LPBE neurons do not project to the medulla, but KF neurons do, this latter projection likely comes from the KF neurons of clusters at1_10 or at1_11, a hypothesis that can now be tested as identifying distinct genetic markers will allow selective genetic access to these populations. Furthermore, we confirmed *Calca*+ neuron types of the KF/PB and their markers in an independent scRNAseq atlas of the same region[38], and tested the correspondence of its clusters with clusters of MERFISH atlases 1-2 (KF/PB) by using Meta-Neighbor, an unsupervised replication framework that employs neighbor voting to quantify the degree of cluster similarity across datasets (Supplementary Fig. 8a–e; Supplementary Data 12, 13; methods)[39,40]. Specifically, scRNA-seq clusters 15 and 16 matched with our MERFISH clusters at2_2 (AUROC = 0.94) and at1_11

(AUROC = 0.87) and were distinguished by the same genes previously identified by our analysis (Fig. 3h; Supplementary Fig. 8d–g; Supplementary Data 11, 14).

## MERFISH-resolved atlas of the PB

The other two divisions of the parabrachial complex are LPB and MPB[32]. To build a transcriptional atlas of the PB, first, we bilaterally traced its boundaries on MERFISH coronal sections spanning from −4.95 to −5.7 bregma level, and then we clustered the 79,413 neurons located within the PB boundaries (methods). The analysis identified 43 clusters, of which 36 belong to the PB. Each cluster was defined by unique gene expression and spatial patterns (Fig. 4a–d; Supplementary Data 15).

Next, we aimed to compare PB neuron types identified by this study with those described in the literature. We observed four different scenarios. (1) Neuron types whose location and marker gene have a correlate in our data: these would include cluster at2_2, *Calca*+/*Il20ra*+, which corresponds with the well-studied CGRP neurons in the LPBE, involved in the response to aversive stimuli[19]; cluster at2_5, *Foxp2*+/*Pdyn*+, which corresponds with dynorphin neurons located in the LPBD, involved in thermoregulation[20]; cluster at2_42, *Nps*+/*Scn5a*+[34]; and cluster at2_13, *Satb2*+/*Col14a1*+, which correspond to *Satb2* neurons located predominantly in the MPB, involved in taste perception[41]. (2) Neuron types with identified location but whose marker gene has not been identified yet: these would include the correspondence of the *Foxp2*+/*Slc32a1*+ population in the MPBE[42] with GABAergic cluster at2_11, which also expresses *Foxp2* but is marked more selectively by *Skor2* and *Gm47757*. This cluster differs substantially from another GABAergic population, cluster at2_16, which is *Foxp2*− and it is marked by *Slc6a5* and *Pax2*, which are expressed at high levels exclusively in the KF and at low levels in the MPBE (Fig. 3f; Supplementary Fig. 9a–d; Supplementary Data 16). Another case is cluster at2_9, *Rxfp1*+/*Runx1*+, which likely corresponds to *Cck*+ neurons in the LPBS projecting to the ventromedial nucleus of the hypothalamus (VMH) and that are responsible for the control of counterregulatory responses to hypoglycemia[43]. (3) Neuron types reported in the literature but with no correlate in our study: these would include *Oxtr*+ cells, which regulate fluid intake[44]; *Tacr1*+ cells, which regulate pain[7,8]; a *Pdyn*+ population, which relays visceral and mechanosensory signals essential for meal termination[16] and a *Foxp2*+/*Pdyn*- cluster, located in an area that Geerling and colleagues called the rostral-to-external Lateral PB subnucleus (PBreL) that is activated at 4 °C, as opposed to a *Foxp2*+/*Pdyn*+ population of the LPBD (cluster at2_5) that is activated at 36 °C[20,45]. Because these genes have been chosen for their correlation with a physiologically activated population of neurons, they might be co-expressed by more than one cluster rather than defining a single neuron type. (4) Neuron types and their marker genes that haven't previously been described in the literature: these would include neurons located in the LPB but especially in the MPB, where only a *Satb2*+ neuron type was previously characterized (Supplementary Fig. 9e, f; Supplementary Data 17)[41]. As assay validation and an example of an uncharacterized population, we confirmed the spatial distribution of cluster at2_26 using RNA-scope staining (Supplementary Fig. 10a–c; Supplementary Data 18, 19). Its neurons express *Foxp2*/*Gpr101* (and *Trhr*) and are located in the part of the PB complex where Kaur et al. have found *Foxp2*+/*Calca*− neurons expressing *cFos* after animals are exposed to high $CO_2$ and that project to respiratory areas of the medulla[46]. This population, located adjacent to cluster at2_2 (*Calca*+/*Il20ra*+), possibly corresponds to cluster at2_26. Identifying *Gpr101* and *Trhr* as markers for these neurons will permit genetic access to them for future investigation.

Finally, we asked if a large population, such as cluster at2_2 (*Calca*+/*Il20ra*+), could harbor transcriptionally defined subpopulations. To test this hypothesis, we isolated all 4,504 neurons from cluster at2_2,

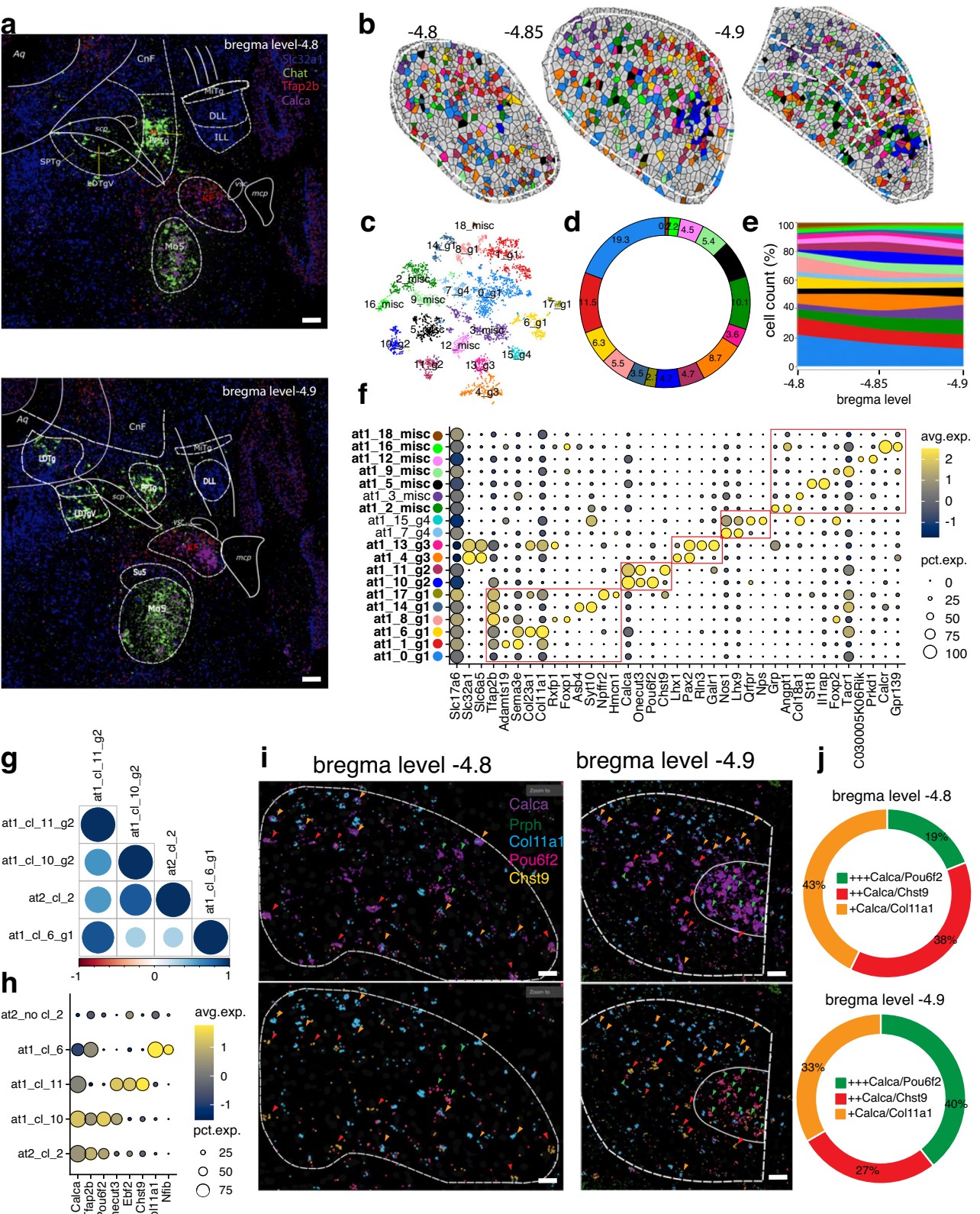

reran them through the same bioinformatic pipeline, and plotted the resulting cells using Voronoi plots. Interestingly, each of the ten *Calca*+ clusters was distinguished by different markers and had a specific spatial pattern (Fig. 4e–h; Supplementary Data 20). Clusters 4 (*Slc6a2*+) and 8 (*Qrfpr*+) were notable: the first is located in the dorsal part of the main cluster at2_2, whereas the second is in its ventral part (Fig. 4i–j). Given the unique transcriptional profiles and spatial

localizations of different subsets of *Calca* neurons, it is interesting to speculate whether these subsets subserve different functions and/or have different afferent and efferent connectivities. For example, in rats, the respiratory parts of the nucleus of the solitary tract project to the rostral ventral portion of the *Calca* territory, gustatory inputs of the caudal ventral part, and gastrointestinal inputs to the middle, dorsal portion[47].

**Fig. 3 | Spatially resolved neuronal atlas of the KF. a** Overlay of Franklin-Paxinos atlas anatomic boundaries on MERFISH image depicting *Slc32a1, Chat, Tfap2b*, and *Calca* transcripts. (scale bar: 200 µm). **b** Voronoi plots depicting KF cells across 3 bregma levels. Glia/non-neuronal cells are in gray. **c** t-SNE plot of 4554 neurons from the KF. **d** Donut plot showing the fraction (%) of each neuronal cluster of the KF. **e** Stacked area chart showing each cluster's cell frequency (cluster trajectory) across 3 bregma levels. **f** Dot plot showing the top 3 markers for each cluster. Red boxes indicate the 5 groups. Bold characters indicate KF clusters; other clusters are from neighboring regions. Clusters displayed by the Voronoi, t-SNE, donut plot, and stacked area chart are color-coded according to the legend in (**f**). **g** Heatmap depicting the Pearson's *r* correlation coefficient of the average expression of 315 genes for all possible combinations of the PB cluster at2_2, KF clusters at1_6, at1_10,

and at1_11. **h** Dot Plot of marker genes specific for PB cluster at2_2, all PB clusters except at2_2, KF clusters at1_6, at1_10, and at1_11. **i** MERFISH images depicting *Calca, Prph, Col11a1, Pou6f2*, and *Chst9* transcripts in the KF at bregma levels −4.8 and −4.9. Green, red, and orange arrows represent high Calca (*Calca+++/Pou6f2*, KF cluster at1_10), medium Calca (*Calca++/Chst9*, KF cluster at1_11), and low Calca (*Calca+/Col11a1*, KF cluster at1_6) neuronal clusters, respectively. (scale bar: 50 µm). **j** Donut plot depicting the fraction of *Calca+* neuronal clusters (clusters at1_6, at1_10, and at1_11) at bregma level −4.8 and −4.9 of the KF. The cluster percentage in plots refers to the images in (**i**). All differentially expressed genes in the dot plot have an average log fold-change >0.25 and an adjusted *p*-value <0.01. Test used: *Wilcoxon Rank Sum two-sided Bonferroni-corrected Test*. Source Data are provided as Source Data file.

## MERFISH-resolved atlas of the MTN, pre-LC, LC, and Bar

To build a transcriptional atlas of an ROI that includes MTN, LC, pre-LC, and Bar, first, we bilaterally traced its boundaries on MERFISH coronal sections spanning from −5.2 to −5.8 bregma level, and then we clustered the 22,358 neurons within the ROI boundaries (methods). Overall, we detected 32 clusters, of which only 27 correspond to neurons of this ROI. Each cluster was characterized by unique gene expression and spatial patterns (Fig. 5a–d; Supplementary Data 21).

The MTN is a paired structure located at the mesopontine junction, which consists of two populations of primary proprioceptive trigeminal sensory neurons that ipsilaterally innervate spindles in the jaw-closing muscles (*first population*; 80–90% of all MTN neurons) or periodontal pressure receptors (*second population*; 10–20% of all MTN neurons)[48]. We identified clusters at3_8 and at3_24 as MTN neurons because of their unique spatial organization and the expression of *Prph, Slc17a7*, and *Pvalb* (Fig. 5e)[13]. Most probably, cluster at3_8 (79% of all MTN neurons) corresponds to the jaw muscle population, whereas cluster at3_24 (21% of all MTN neurons) to the periodontal one. We also identified unique marker genes for cluster at3_8 versus at3_24 (adj. *p*-value < 0.01) (Fig. 5e; Supplementary Data 22) that will allow studying their different properties.

The LC is the primary source of noradrenergic innervation of the cerebral cortex and cerebellum, and it is located in the dorsolateral PnTg on the lateral floor of the fourth ventricle[3]. It receives input from widespread brain regions and projects throughout the forebrain, brainstem, cerebellum, and spinal cord[3]. Recently, it has been demonstrated that the modular input-output organization of the LC can enable temporary, task-specific modulation of different brain regions[3]. However, whether this modularity corresponds to transcriptionally defined groups of noradrenergic neurons is still undetermined. To this end, we isolated 4074 noradrenergic neurons from cluster at3_0 and reran them through the same bioinformatic pipeline. Each cluster was distinguished by different markers and had a specific spatial pattern (Fig. 5g–j; Supplementary Data 23): clusters 0, 1, 4, and 5 were distributed across the LC, whereas clusters 2 (*Col18a1+/Gpr101+*) and 3 (*Tacr3+/Ecel1+*) were located in the dorsal portion of the caudal LC and the ventral part of the rostral LC, respectively (Fig. 5g, i). As cortical projections arise mainly from the dorsal LC and spinal projections from the ventral LC, it would be interesting to determine whether these populations have different targets[49,50]. In addition, we report two LC non-noradrenergic populations: a low-expressing *Slc17a6/Slc32a1* population likely corresponding to cluster at3_10 and a population of *Penk* neurons that is part of cluster at3_1 (Fig. 5d).

The term "pre-locus coeruleus" broadly refers to a small region that lies on both sides of the LC, approximately from bregma levels −5.3 to −5.7. It was initially coined by Geerling and colleagues to identify a neuronal population located ventromedial to the rostral LC that receives excitatory inputs from aldosterone-sensing HSD2 neurons of the NTS[51,52], expresses *Foxp2* and *Pdyn* genes[53], and has elevated levels of *cFos* during dietary sodium deprivation[54]. Our analysis detected six clusters restricted to the pre-LC (group 1 except clusters

at3_0, at3_8, and at3_24) and another seven whose cells were shared with medial regions (group 3) (Fig. 5a, b). Based upon their gene expression, these neurons discovered by Geerling and colleagues could correspond to cluster at3_30, *Tnc+/Rxfp2+*. Notably, cluster at2_5 from the PB also expresses *Foxp2, Pdyn*, and *Th*, and its cells project to the preoptic area (PoA) and hypothalamus[53,55]. DE analysis identified the top 5 (adj. *p*-value < 0.01) marker genes for cluster at3_30. Of note, none of the genes in our MERFISH panel was a marker for PB cluster at2_5 (Fig. 5f; Supplementary Data 24).

The Bar is a small nucleus located between the LC and the LDTg and is critical for bladder voiding[56]. Bar neurons send long-range projections to the lumbosacral level of the spinal cord, where bladder- and external urethral sphincter-innervating motor neurons reside[18]. While more than half of the Bar neurons express *Crh*[57] and stimulation of Bar^Crh neurons promotes bladder contractions[58], other Bar neurons' genetic and functional identity remains elusive. Our analysis detected nine clusters in the "medial region" (group 2). Crh-expressing cluster at3_2 is the main glutamatergic population (Fig. 5a–c). Other glutamatergic clusters in or near Bar include at3_9 (*Lhx4+/Vsx2+*), at3_26 (*Vglut3+*), and at3_1 (*Penk+/Mc4r+*) (Fig. 5a–d). It was recently shown that photo-inhibition of Bar^Esr1 neurons terminates ongoing urethral sphincter relaxation and stops voiding[59]. We detected *Esr1* transcript in cluster at3_2 (*Crh+*) and, to a lesser extent, in neurons of other Bar clusters (Supplementary Fig. 9d). Among the GABAergic populations, clusters at3_12 (*Crhbp+/Glp1r+*) and at3_7 (*Lhx1+/Gm47757+*) are intermingled with the *Crh+* neurons of Bar, and GABAergic clusters at3_25, at3_27, at3_28, and at3_31 surround the nucleus. These inhibitory populations could represent neurons in the CGPn or local interneurons that influence Bar's neuronal activity[60]. Furthermore, Bar neurons have extensive dendritic arbors[18], and cholinergic neurons in LDTg (cluster at3_29) are likely in close contact with the *Crh+* neurites[61].

## MERFISH-resolved atlas of the brain nuclei of the medial part of the dPnTg

To investigate a ROI that includes LDTg, VTg, DTg, CGA, CGB, Sph, O, and CGPn, we first traced its boundaries on MERFISH coronal sections spanning from bregma level −4.7 to −5.8, and then we clustered the resulting 120,182 neurons within the ROI boundaries (methods). Overall, we detected 46 clusters, of which only 38 corresponded to neuronal types within this ROI. Each cluster was characterized by unique gene expression and spatial patterns (Fig. 6a–d; Supplementary Data 25).

The LDTg borders the LC and the DTg through some of its course from bregma level −4.7 to −5.6[62]. To decipher its spatial organization, we first computed the contribution of each cluster to the LDTg/LDTgV region and then its trajectory across 11 rostrocaudal levels (Fig. 7b, c, h). The analysis detected 27 LDTg/LDTgV neuronal clusters, of which 17 (55%) are GABAergic, 8 (32%) glutamatergic, and 2 (13%) cholinergic (Fig. 7b). Strikingly, similar ratios were documented by Luquin E. et al. in rats (Fig. 7d, left side)[63]. In our analysis, the well-characterized cholinergic population of the LDTg corresponded to clusters at4_1 and

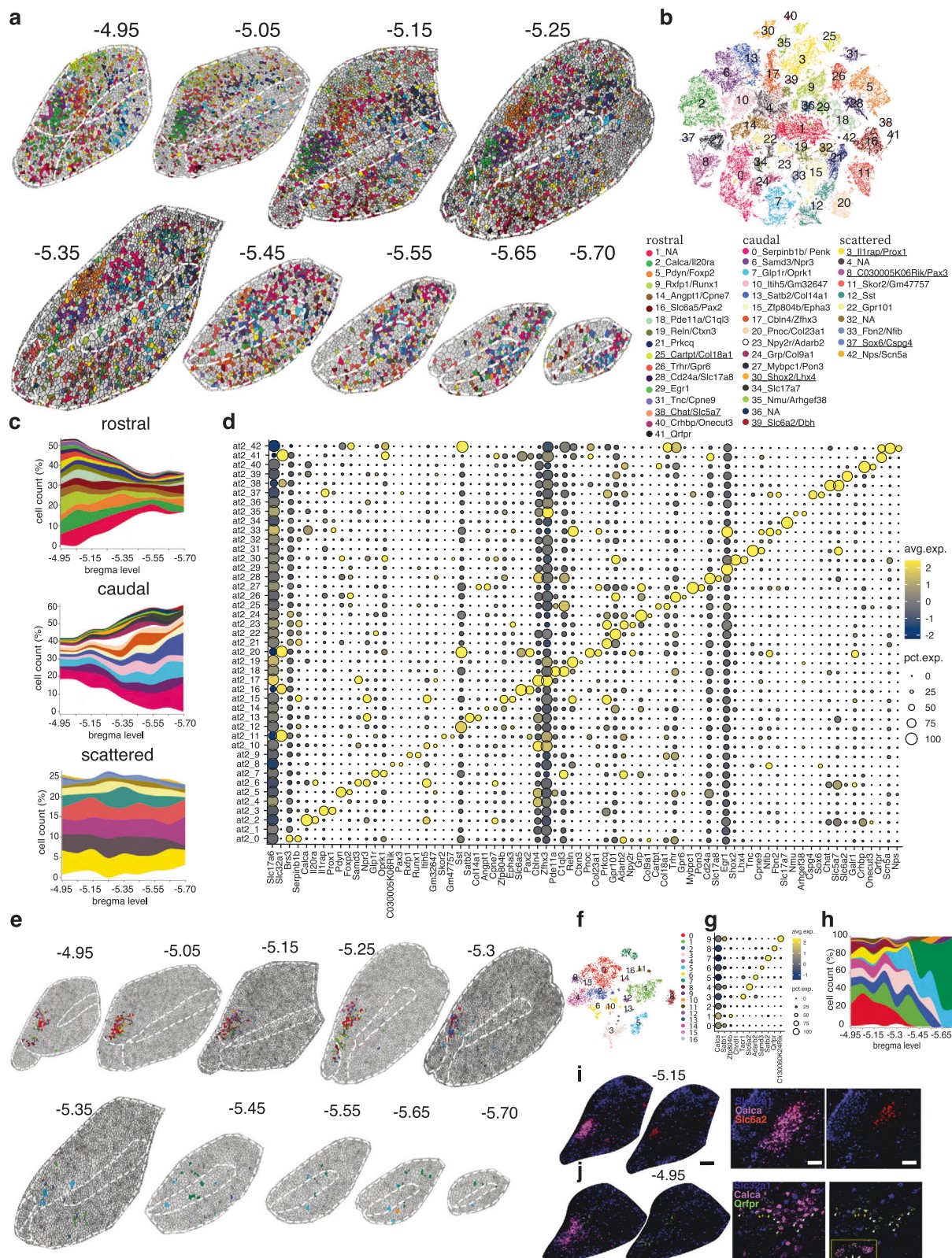

at4_39. These neurons are active during wakefulness and REM sleep[64], and cannot release glutamate or GABA[63], lacking *Slc17a6* and *Slc32a1* expression (Fig. 6d). Glutamatergic neurons of the LDTg region are mainly represented by *Shox2*+ clusters at4_0 and at4_7 (also *Lhx4*+), whose cells are uniformly distributed from rostral to caudal, where they gradually replace the cholinergic neurons. The remaining glutamatergic clusters are primarily rostral (Fig. 7b, c). Among them, cluster

at4_28 (*Tnc*+) is only found in the LDTgV, representing a potential marker to study its specific function[65] (Fig. 7a). Conversely, GABAergic clusters showed a more specific spatial distribution along the rostrocaudal axis (Fig. 7b, c). Finally, previous work has identified a population of *Glp1r*+ neurons in the LDTg that play a role in attenuating cocaine-seeking behavior by projecting to the ventral tegmental area (VTA)[10]. In our dataset, *Glp1r*+ neurons of the LDTg corresponded

**Fig. 4 | Spatially resolved neuronal atlas of the PB. a** Voronoi plots depicting PB cells across 9 sequential MERFISH sections from −4.95 to −5.7 bregma level. Glia/non-neuronal cells are in gray. **b** t-SNE plot of 79,413 neurons. **c** Stacked area charts showing each cluster's cell frequency (cluster trajectory) across all 9 bregma levels. Clusters displayed by the Voronoi, t-SNE, and stacked area chart are color-coded according to the legend in (**b**). Clusters underlined in the legend represent external-to-the-PB neuron types/glia contamination. **d** Dot plot of the top 2 markers for each cluster. **e** Voronoi plots representing neurons from PB cluster at2_2 across 10 sequential coronal sections from bregma level −4.95 to −5.7. Other PB neuronal clusters and glia/non-neuronal cells are in gray. **f** t-SNE plot representing 4504 neurons. **g** Dot plot depicting the *Calca* gene and the top marker for each *Calca*+ subcluster. **h** Stacked area chart showing the cluster trajectory across the 10 sequential MERFISH sections in (**e**). Clusters displayed by the Voronoi, t-SNE, and stacked area chart are color-coded according to the legend in (**f**). **i** Left: MERFISH image of *Slc32a1*, *Calca*, and *Slc6a2* transcripts in the PB complex at bregma level −5.15. (scale bar: 200 μm). Right: enlarged view of the *Calca*+ cluster 4. (scale bar: 100 μm). **j** Left: MERFISH image of *Slc32a1*, *Calca*, and *Qrfpr* transcripts in the PB complex at bregma level −4.95. (scale bar: 200 μm). Right: enlarged view of the *Calca*+ cluster 8. (scale bar: 50 μm). In (**g**, **h**), only clusters composed of >100 cells were included. All differentially expressed genes in the dot plot have an average log fold-change >0.25 and an adjusted *p*-value <0.01. Test used: *Wilcoxon Rank Sum two-sided Bonferroni-corrected Test*; NA, no marker detected.

to GABAergic clusters at4_21 and at4_38 (-68%) and the glutamatergic cluster at4_22 (-32%) (Fig. 7d, right side)[10].

Gudden's tegmental nuclei comprise the VTg and DTg. In the rat, both divisions send heavy projections to the mamillary bodies: the VTg innervates the medial mammillary nucleus, supporting spatial learning, whereas the DTg innervates the lateral mammillary nucleus, supporting spatial navigation[66]. The VTg is located near the midline from −4.7 to −5.2 bregma level and is a purely GABAergic nucleus (Fig. 7e, f). In fact, >90% of its neurons belong to GABAergic cluster at4_6 (*Satb1*+), while the remaining are from clusters at4_4 (*Tacr1*+), at4_38 (*Robo3*+) and at4_40 (*Calca*+) (Fig. 7f). The DTg is also located near the midline, from −5 to −5.8 bregma level, and it is composed of three divisions: the DTgP, DTgC, and PDTg (Fig. 7h). Its primary function is in landmark and directional navigation, and its cells, referred to as head direction (HD) cells, fire in response to changes in head velocity and direction (i.e., left, right)[12]. To decrypt its spatial organization, we first computed the overall contribution of each cluster to the DTg and then its trajectory across ten sequential rostrocaudal levels (Fig. 7g, i). Our analysis identified 21 clusters divided into GABAergic (17/21 clusters, representing 88.5% of DTg neurons) and glutamatergic (4/21 clusters, representing 11.5% of DTg neurons) (Fig. 7g). Next, we investigated their spatial location with respect to their anatomical organization. The DTgP extends from bregma level −5 to −5.6 (Fig. 7h). The rostral-central part of the DTgP is mainly characterized by GABAergic clusters at4_3 (*Vmn1r209*+), at4_5 (*Gpr39*+), at4_12 (*Nts*+), and at4_16 (*Onecut1*+) (Fig. 7i, k). In contrast, its caudal part is mostly glutamatergic; it harbors cluster at4_0 (*Shox2*+) and a small GABAergic *Npy*+ population corresponding to cluster at4_36 (Fig. 7i, k). The DTgC borders the DTgP to its extent, except in its rostral part (Fig. 7h). The rostral-central part of the DTgC is populated exclusively by the GABAergic cluster at4_4 (*Tacr1*+) that ends in the caudal region, intermingled with the glutamatergic cluster at4_25 (*Lhx9*+) (Fig. 7k). Finally, the PDTg occupies the very caudal portion of the DTg, from −5.7 to −5.8 bregma level (Fig. 7h); it represents a point where VTg, DTgP, DTgC, and the Sph, which is dorsal to the DTgP from −5.4 to −5.55 bregma level, converge into one structure. (Fig. 7k). Our analyses indicated that the Sph is composed of >90% of GABAergic neurons belonging to clusters at4_18 (*Ebf2*+) and at4_41 (*Rxfp1*+) (Fig. 7j).

Finally, we examined the nucleus O (also known as nucleus incertus (NI)), CGA, and CGB. The NI extends from −5.3 to −5.6 bregma level and consists of a midline, bilateral cluster of large, multipolar neurons in the central gray[67] (Fig. 7h; Supplementary Fig. 11c). Recent evidence suggests its involvement in modulating arousal, feeding, stress responses, anxiety, addiction, attention, and memory by projecting to high-order structures of the forebrain[4,5,14]. Despite its main GABAergic population being known to express *Rln3*, the genetic makeup of the other neuronal subtypes is unknown. As before, we first computed the overall contribution of each cluster to the NI, CGA, and CGB brain nuclei and then its trajectory across five sequential rostrocaudal levels (Supplementary Fig. 11a–c). Our analysis identified 24 neuronal clusters, of which 16 (58.2%) are GABAergic and 8 (41.8%) are

glutamatergic (Supplementary Fig. 11a). None of these neuron types was previously documented.

## Correspondence between MERFISH and snRNA-seq neuronal clusters of the dPnTg allows whole transcriptome imputation

Given the limited number of genes profiled by MERFISH, we sought to determine the degree to which neuronal clusters identified by MERFISH in the four subregions corresponded to snRNA-seq clusters. This would allow the transfer of transcriptional and spatial information between the two datasets. To this end, we applied MetaNeighbor[39,40] (methods). We found that 94/114 MERFISH-identified clusters corresponded to 82/127 snRNA-seq-identified clusters, and this correspondence was reciprocal in 50/122 instances (AUROC > 0.85; Fig. 8a, b; Supplementary Data 26; methods). While only for "mutual" matches, i.e., those having 1:1 correspondence, it is possible to directly infer the expression of genes not probed by MERFISH from the snRNAseq dataset, "non-mutual" correspondences are still useful because they help restrict the field of investigation. Of note, "non-mutual" or missing matches between clusters of the two datasets could stem from the difference in technology sensitivity, number of neurons profiled, features used, and difference in contamination from neighboring regions due to precision in dissecting the ROI.

## Comparison between mouse and human neuronal subtypes reveals a high degree of transcriptional similarity

A recent publication[68] made snRNA-seq data from the human pons accessible. We retrieved and pooled together all the nuclei from two dissection biopsies: the first including the pontine reticular formation (PnRF) and the PB; the second, the DTg and all other medial nuclei of the dPnTg. A pre-filtered dataset of 50,250 high-quality nuclei x 37,165 genes was analyzed by using our bioinformatic pipeline (methods)[27-30]. The analysis identified 32 clusters that we grouped into 10 main cell types (Supplementary Fig. 12a, b; Supplementary Fig. 13a, b, e; Supplementary Data 27). Next, we isolated only the neurons, excluding the glial/non-neuronal clusters, and divided them into two main groups for re-clustering. The "excitatory neurons" group included 17,995 nuclei divided into 38 clusters, whereas the second group, "inhibitory neurons", included 11,871 nuclei divided into 29 clusters (Supplementary Fig. 13c, d). Each cluster was defined by the expression of one or a combination of marker genes (Supplementary Fig. 13f, g; Supplementary Data 28, 29). Albeit to a different extent, every covariate contributed to each neuronal cluster (% of cells), confirming the mitigation of the batch effects (Supplementary Fig. 12c–g).

Next, given the extensive use of *Mus Musculus* as a model to study neuronal circuits, we employed MetaNeighbor to evaluate the inter-species degree of transcriptional similarity (methods). Interestingly, 50/67 human snRNA-seq clusters corresponded to 52/127 mouse snRNA-seq clusters, and this correspondence was mutual in 23/64 instances, indicating a medium-high interspecies transcriptional similarity (Supplementary Fig. 13h; Supplementary Data 30). This fact could underlie an evolutionarily conserved function of this brain

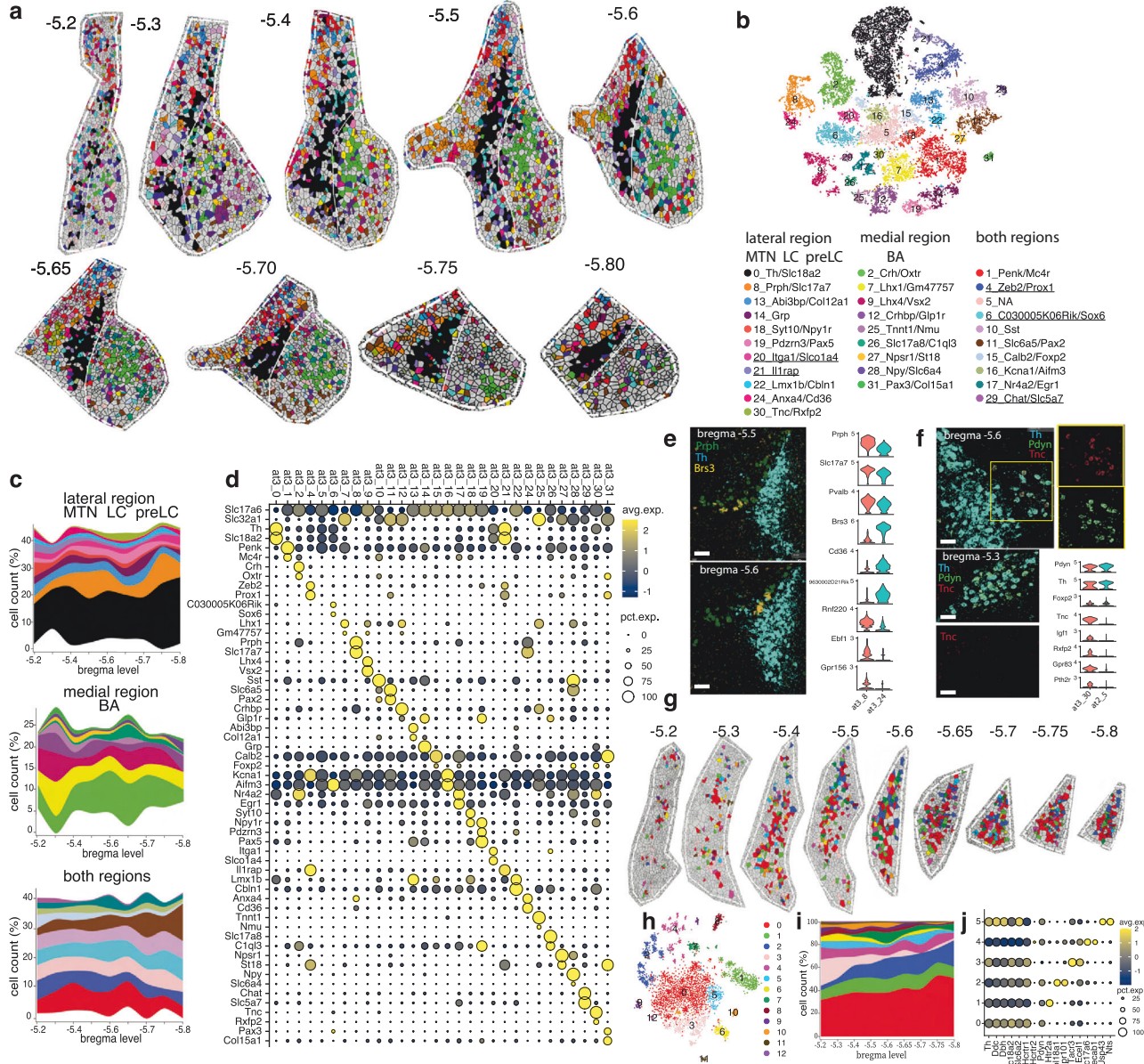

**Fig. 5 | Spatially resolved neuronal atlas of the MTN, pre-LC, LC, and Bar.**
**a** Voronoi plots depicting cells of a ROI that includes MTN, pre-LC, LC, and Bar across 9 sequential sections, from −5.2 to −5.8 bregma level. Glia/non-neuronal cells are in gray. **b** t-SNE plot of 22,358 neurons. **c** Stacked area chart showing each cluster's cell frequency (cluster trajectory) across all 9 MERFISH sections. Clusters displayed by the Voronoi, t-SNE, and stacked area chart are color-coded according to the legend in (**b**). Clusters underlined in the legend represent external-to-the-ROI neuron types/glia contamination. **d** Dot plot of the top 2 markers for each cluster. **e** Left: MERFISH image showing the spatial distribution of *Prph*, *Th*, and *Brs3* at bregma levels −5.5 and −5.6. (scale bar: 75 μm). Right: violin plots depicting the average expression level (*y*-axis) of 9 genes in clusters at3_8 and at3_24 (*x*-axis). **f** Left: MERFISH image showing the spatial distribution for *Th*, *Pdyn*, and *Tnc* in bregma levels −5.6 and −5.3 in the pre-LC (top) and LPBD (bottom). (scale bar: 50

μm). Right: violin plots depicting the expression level (*y*-axis) of 8 genes in clusters at3_30 (pre-LC) and at2_5 (LPBD) (*x*-axis). **g** Voronoi plots depicting LC nora-drenergic neurons across 9 sequential MERFISH sections from −5.2 to −5.8 bregma level. Other ROI's neuronal clusters and glia/non-neuronal cells are in gray. **h** t-SNE plot of 4,074 noradrenergic neurons. **i** Stacked area chart showing each cluster's cell frequency (cluster trajectory) across 9 sequential MERFISH sections in (**g**). Clusters displayed by the Voronoi, t-SNE, and stacked area chart are color-coded according to the legend in (**h**). **j** Dot plot depicting *Th*, *Ddc*, *Dbh*, *Slc18a2*, *Slc6a2*, *Hcrtr1*, and *Hcrtr2* genes and the top 2 marker genes for each subcluster. In (**i**, **j**), only clusters composed of >200 cells were included. All differentially expressed genes in the dot plot have an average log fold-change >0.25 and an adjusted *p*-value <0.01. Test used: *Wilcoxon Rank Sum two-sided Bonferroni-corrected Test*; NA, no marker detected.

region. Finally, to gain more insights into the functional relationships of genes driving cell-type replicability, we applied a supervised version of MetaNeighbor that uses clusters with "reciprocal" matches and tests a list of gene sets. We used the *Mus Musculus* gene ontology (GO) (methods) as gene sets. The top-scoring average AUROCs pinpointed GO terms related to neurotransmitters/synaptic functions and neuro-peptides, meaning these GO gene sets are moderately conserved functional gene ensembles contributing to cell-type replicability

between the two species (Supplementary Fig. 14e, f; Supplementary Data 31).

Finally, we decided to focus on the PB because it has shared anatomy between humans and mice and because scRNA-seq data from PB are publicly available for both species[38,68]. After discarding GABAergic, cholinergic, serotoninergic, and noradrenergic clusters to limit the contamination from neighboring areas, a dataset of 6,638 putative human PB glutamatergic neuronal nuclei was re-clustered.

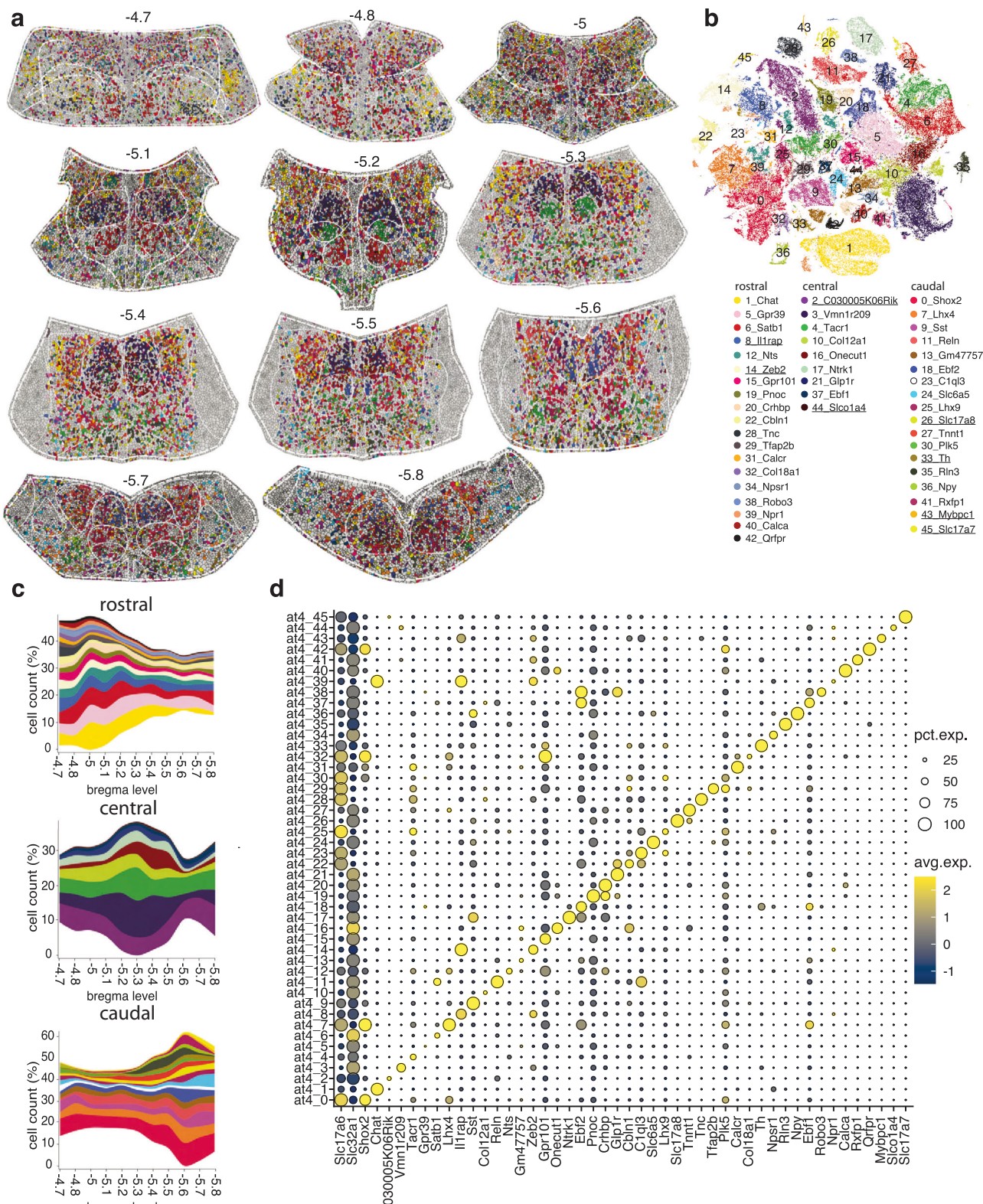

**Fig. 6 | Spatially resolved neuronal atlas of LDTg, DTg, VTg, Sph, NI, CGA, CGB, and CGPn. a** Voronoi plots depicting cells of a ROI that includes LDTg, DTg, VTg, Sph, NI, CGA, CGB, and CGPn across 11 sequential sections from −4.7 to −5.8 bregma level. Glia/non-neuronal cells are in gray. **b** t-SNE plot of 120,182 neurons. **c** Stacked area charts showing each cluster's cell frequency (cluster trajectory) across all 11 MERFISH sections. Clusters displayed by the Voronoi, t-SNE, and stacked area chart are color-coded according to the legend in (**b**). Clusters underlined in the legend represent external-to-the-ROI neuron types/glia contamination. **d** Dot plot of the top marker for each cluster. All differentially expressed genes in the dot plot have an average log fold-change >0.25 and an adjusted *p*-value <0.01. Test used: *Wilcoxon Rank Sum two-sided Bonferroni-corrected Test*. NA, no marker detected.

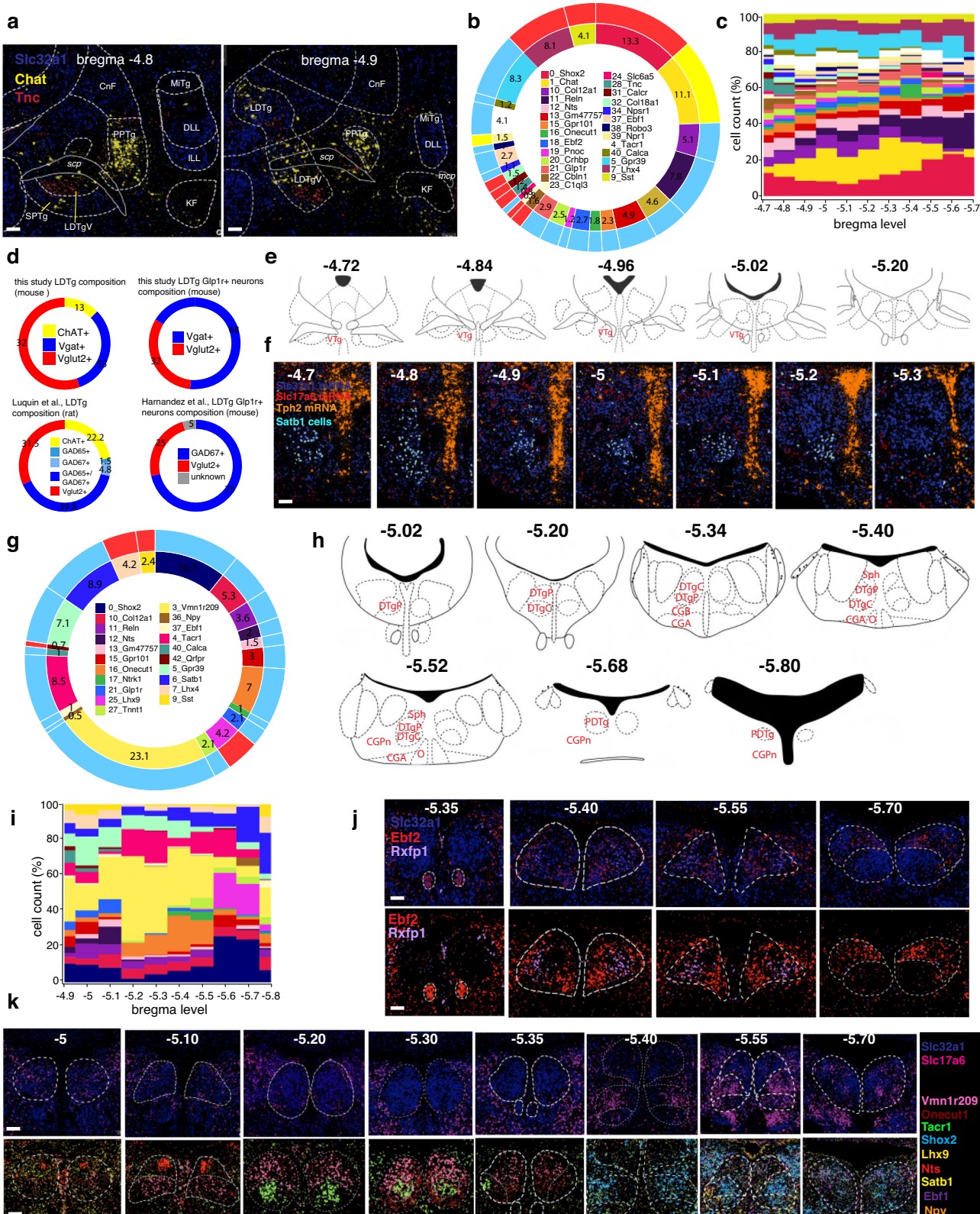

The analysis identified 36 clusters specified by distinct marker genes (Supplementary Fig. 14a, b; Supplementary Data 32). MetaNeighbor analysis run between mouse PB scRNA-seq atlas[38] versus human snRNA-seq PB atlas revealed many clusters had a high degree of transcriptional similarity between the two species. 29/36 human snRNA-seq clusters corresponded to 20/21 mouse scRNA-seq clusters, and this correspondence was mutual in 15/33 instances (Supplementary Fig. 14c, d; Supplementary Data 33). Of note, the anatomy of human clusters 14 (*CALCA*+/*ALCB*+) and 21 (*NPS*+/*FOXP2*+) have also been confirmed by immunohistochemistry in sections of human post-mortem brain tissue to be homologous to those in rodents[69,70]. For cluster 4, the human *CGRP* cell group is in the exact relative location as the LPB *Calca* neurons in mice, and *CGRP* terminals were found in the same forebrain areas targeted by *CGRP* neurons in rodents[35,69].

**Fig. 7 | In-depth characterization of the LDTg, VTg, DTg, and Sph. a** Overlay of Franklin-Paxinos atlas anatomic boundaries on MERFISH image depicting *Slc32a1*, *Chat*, and *Tnc* transcripts. (scale bar: 250 μm). **b** Donut plots: the inner plot shows the overall contribution (%) of each cluster to the total LDTg/LDTgV neurons; the outer plot classifies the clusters as glutamatergic (red), GABAergic (light blue) and cholinergic (yellow). **c** Stacked area charts of the LDTg/ LDTgV cluster trajectory. Clusters are color-coded according to the legend in (**b**). **d** Left: donut plot showing the LDTg cell partition in glutamatergic (red), GABAergic (blue), and cholinergic (yellow) in this study and as reported by Luquin et al. Right: estimation of *Glp1r* +/*Slc32a1*+ and *Glp1r*+/*Slc17a6*+ cells in mouse LDTg by this study and as reported by Hernandez et al. **e** Schematic from the Paxinos atlas showing the VTg anatomical location. **f** MERFISH image showing cluster at4_6 (VTg neurons; cyan polygons) along with *Slc32a1*, *Slc17a6*, and *Tph2* transcripts. (scale bar: 100 μm). **g** Donut

plots: the inner plot shows the overall contribution (%) of each cluster to the total DTg neurons; the outer plot classifies the clusters as glutamatergic (red) and GABAergic (light blue). **h** Schematic from the Franklin-Paxinos atlas showing the DTg, Sph, NI, CGA, and CGB anatomical location from −5.02 to −5.8 bregma level. For (**e**, **h**), abbreviations refer to Table 1. **i** Stacked area charts of the DTg cluster trajectory. Clusters are color-coded according to the legend in (**g**). **j** Overlay of Franklin-Paxinos atlas anatomic boundaries on MERFISH images depicting *Slc32a1*, *Ebf2*, and *Rfxfp1* (top) and *Ebf2* and *Rfxp1* marker genes (bottom) in the Sph. (scale bar: 100 μm). **k** Overlay of Franklin-Paxinos atlas anatomic boundaries on MERFISH images depicting *Slc32a1* and *Slc17a6* (top) and 9 marker genes (bottom) in the DTg across the same rostrocaudal levels. (scale bar: 100 μm). Legend is on the right side of both panels. In (**b**, **g**), only clusters contributing >0.5 % to the overall neuronal population were plotted.

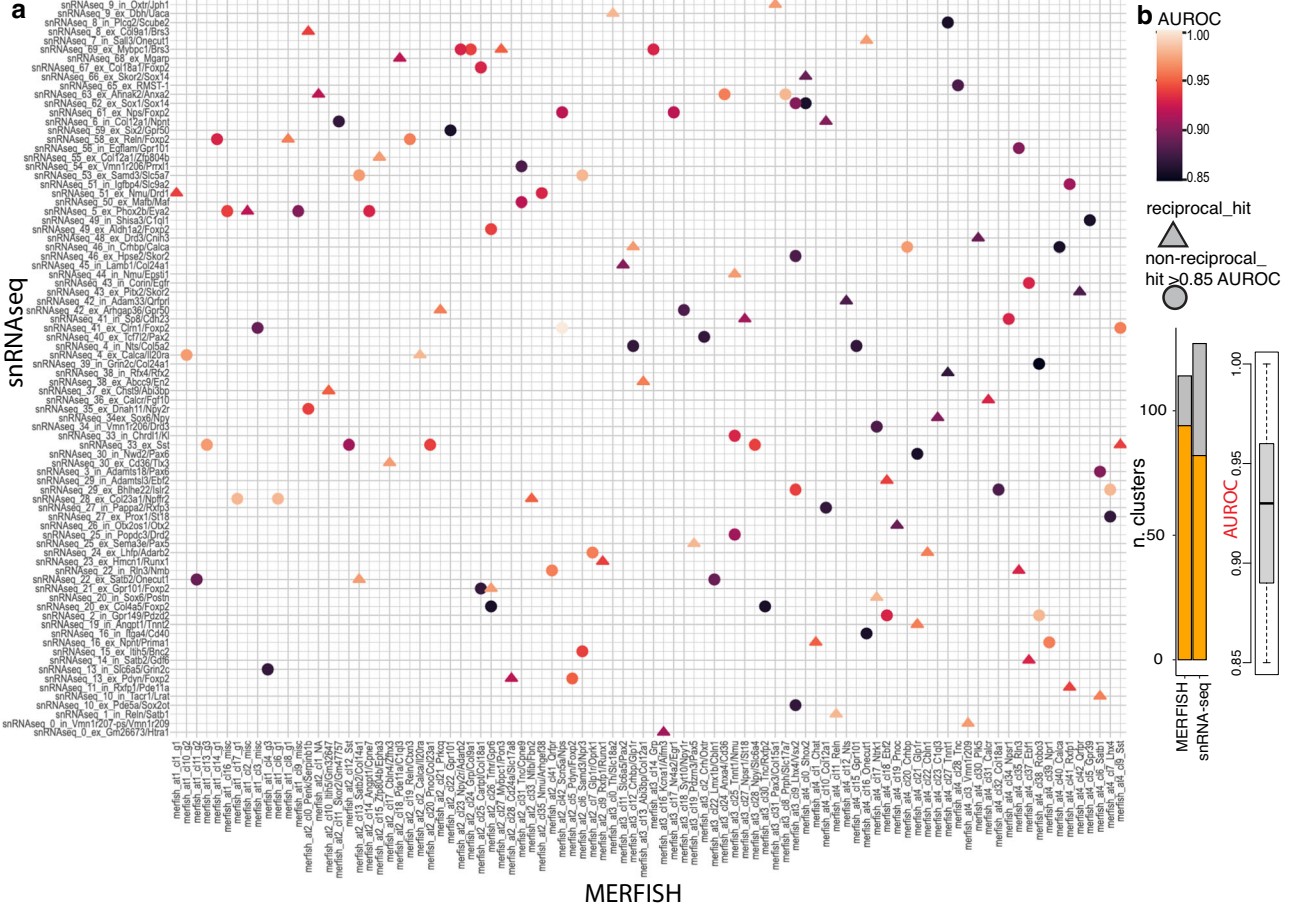

**Fig. 8 | Cluster correspondence between mouse snRNA-seq and MERFISH data. a** Heatmap depicting the cluster correspondence between snRNA-seq (78,485 neuronal nuclei grouped in 127 clusters) and MERFISH atlases 1-4 (193,714 neuronal cells grouped in 114 clusters) datasets of the dPnTg. Legend defining the AUROC score and the "match type" (reciprocal vs non-reciprocal) is on the right side of (**a**). **b** Left: stacked bar plot showing the number of clusters with a match (orange) over the total clusters (gray) identified by MERFISH (atlases 1–4) and snRNA-seq

approaches. Right: boxplot showing the AUROC scores distribution. The black middle line denotes the median value (50th percentile), while the gray box contains the 25th to 75th percentiles of the dataset. The black whiskers mark the 5th and 95th percentiles, and values beyond these upper and lower bounds, marked with black dots, are considered outliers. AUROC, area under the receiver operator characteristic curve; NA, no marker detected; CONT, glial contamination.

## Discussion

To gain selective access and mechanistically investigate the neuronal subtypes within the dPnTg, it is necessary to identify their spatial location and transcriptional identity, particularly their marker genes. While the field presently has characterized some genetic markers for this region, the transcriptional identity of most neuronal subtypes has remained elusive. By combining snRNA-seq and MERFISH, we generated a spatially resolved transcriptional atlas of the dPnTg at a single-

cell resolution. This study analyzed ~1 million cells and identified over 120 neuronal clusters across four anatomical subregions of the dPnTg, confirming the remarkable degree of transcriptional diversity in this region[25,26,68]. To accomplish this, we employed an unsupervised approach, snRNA-seq, to identify the most informative genes and then a supervised approach, MERFISH, relying on a subset of 315 genes, to spatially resolve the neuronal clusters. Finally, we applied Meta-Neighbor, an unsupervised replication framework that employs

neighbor voting to quantify the degree of cluster similarity across MERFISH and snRNA-seq datasets while preserving the dataset independence[39,40]. Mapping the correspondence between clusters using MetaNeighbor allows the transfer of transcriptional and spatial information from one dataset to another.

Overall, we spatially characterized the transcriptome of the mouse dPnTg at single-cell resolution, identified the neuronal subtypes populating this region, spatially located them, and provided the marker genes that specify each subtype. In addition, we related this information to the scientific literature to reconcile our findings with the field's current state of knowledge. Our spatially resolved transcriptional atlas should greatly facilitate future mechanistic investigations of neural circuits in this region. For example, knowing the genetic markers allows for generating recombinase-driver mice that can be used to access specific neuronal populations to perform behavioral, neuronal tracing, and activity mapping experiments[71]. Furthermore, to grant the scientific community easy access to this resource, we developed a GPU-powered visualizer (http://harvard.heavy.ai:6273/) to query the dPnTg MERFISH-resolved datasets, which includes a representative series of 12 sequential coronal sections cut at intervals of 80-90 μm that span bregma levels from −4.7 to −5.8. By leveraging these two molecular techniques, we built a spatially resolved transcriptomic atlas of the dPnTg at single-cell resolution and made the dataset accessible and interactive. This will allow future studies to shed light on the function of the many neuronal subtypes populating this region.

## Methods

### Mouse strains and brain dissections
DroNc-seq and MERFISH experiments were performed on C57BL/6J background mice purchased from the Jackson Laboratory (JAX). Mice were housed at 25 °C, -55% humidity, on a 12:12-h light/dark cycle. Animal experiments were approved by the Beth Israel Deaconess Medical Center's Institutional Animal Care and Use Committee (IACUC) (protocol no. 047-2022). A total of 9 and 8 batches (3-5 mice each) of male and female mice, respectively, 8–10 weeks old, were used for DroNc-seq. To obtain a more precise dissection of the dPnTg and minimize the contamination from neighboring areas, such as the cerebellum, we labeled two nuclei that define its extension: the PB and the Bar.

To visualize the PB, we exploited the fact that the PB receives extensive synaptic inputs from the NTS[72]. A Cre-expressing adeno-associated virus, AAV1-hSyn-Cre (pENN-AAV1-hSyn-Cre-WPRE-hGH; titer ≥ 1×10[13] vg/ml; Addgene, 105553), was injected into the NTS of an Ai14 mouse. The Ai14 mouse (JAX, stock no. #007914, Gt(ROSA) 26Sor[tm14(CAG-tdTomato)Hze]) has a Cre reporter allele with a loxP-flanked STOP cassette preventing transcription of a CAG promoter-driven red fluorescent protein variant (tdTomato), all inserted into the Gt(ROSA)26Sor locus. Injection of AAV-Cre into the NTS results in the expression of tdTomato, which travels through the projections from the NTS to label the PB specifically. Two weeks after the AAV injection, mice were decapitated for brain dissection. For micro-dissection of specific brain areas (PB, Bar), mice were rapidly decapitated without anesthesia to avoid any drug-induced effects on transcription. To visualize the Bar, we exploited the highly selective expression of *Crh* in this brain nucleus[18]. Crh-IRES-Cre mice (JAX, stock no. #012704, B6(Cg)-Crh[tm1(cre)Zjh/J]) were crossed with EGFP-L10a (JAX, stock no. #024750, B6;129S4-Gt(ROSA)26Sor[tm9(EGFP/Rpl10a)Amc/J]) to obtain Crh-IRES-Cre::EGFP-L10a mice whose *Crh*-expressing neurons were selectively labeled with GFP.

In both approaches, mice were sacrificed between 10 am and 1 pm. To avoid any stress-related transcriptional changes, mice were decapitated immediately after removal from home cages. After decapitation, the brain was removed from the skull, chilled for 3 min in an ice-cold DMEM/F12, no phenol red (Thermo Fisher Scientific) media slush, and placed ventral surface up in an ice-cold stainless steel brain matrix (Roboz Surgical Instrument Co). A 1 mm thick coronal slice was cut, and the area of interest was dissected bilaterally using a micro dissecting knife (Roboz Surgical Instrument Co.) under the fluorescent stereotactic microscope (Zeiss Discovery V8). Dissections were flash-frozen in dry ice and stored at −80 °C.

### Stereotactic injection into the NTS
Stereotaxic AAV injections into the NTS were performed in seven- to ten-week-old male/female mice under ketamine (100 mg/kg) and xylazine (10 mg/kg) anesthesia. Ketamine and xylazine were diluted in 0.9% sterile isotonic saline and injected into the intraperitoneal cavity. Mice were then placed into a stereotaxic apparatus (David Kopf model 940) with the head angled down at approximately 60°. An incision was made at the level of the cisterna magna, and skin and muscle were retracted to expose the dura mater covering the 4th ventricle. A 28-gauge needle was used to cut through the dura and allow access to the brainstem. Subsequently, a pulled glass micropipette (20–40 mm diameter tip) was used to inject AAV1-hSyn-Cre into the NTS. Stereotaxic coordinates were anterior 0.3 mm, lateral ± 0.15 mm, and ventral 0.3 mm from calamus scriptorius. The virus (200 nl) was injected by an air pressure system using picoliter air puffs through a solenoid valve (Clippard EV 24VDC) pulsed by a Grass S48 stimulator to control injection speed (40 nl/min). The pipette was removed 3 min post-injection, followed by wound closure using absorbable suture for muscle and silk suture for the skin. Subcutaneous injection of sustained-release Meloxicam (4 mg/kg) was provided as postoperative care.

### Nuclei isolation
5-6 bilateral tissue dissections were placed in a dounce homogenizer with 1 mL cold (4 °C) Lysis Buffer containing 10 mM trisHCl pH 8 (Sigma-Aldrich), 250 mM Sucrose (Sigma-Aldrich), 25 mM KCl, 5 mM MgCl₂ (Sigma-Aldrich), 0.1% Triton x100 (Sigma-Aldrich), 0.5% RNasin Plus RNase Inhibitor (Promega), 0.1 mM Dithiothreitol (DTT) (Sigma-Aldrich) in UltraPure™ DNase/RNase-Free Distilled Water (Thermo Fisher Scientific). After douncing for 20 times, the solution was filtered through a sterile 20 μm Cell Strainer (pluriSelect), collected in 1.5 ml DNA LoBind® Tubes (Eppendorf), and centrifuged for 10 min at 900 g (rcf) at 4 °C. The "slow sedimenting" component (debris and membranes) was aspirated and discarded while the "fast sedimenting" component (nuclear fraction) was gently resuspended in a 1 mL of Working Solution containing 1X pH 7.4 RNase free PBS (Thermo Fisher Scientific), 0.01% Albumin Bovine Serum (BSA) (Sigma-Aldrich), 0.5% RNasin Plus RNase inhibitor (Promega) in UltraPure™ DNase/RNase-Free Distilled Water (Thermo Fisher Scientific). Nuclei were kept on ice while transferred to the BNORC Functional Genomics and Bioinformatics (FGB) Core for DroNc-seq assay.

### DroNc-seq assay, library preparation, and sequencing
DroNc-seq-seq was performed as per Habib et al., with minor modifications[23]. Briefly, nuclei stained with Hoechst 33342 (Thermo-Fisher, cat. R37605) were counted on a hemocytometer and diluted in NSB to -250,000 nuclei/ml. Barcoded beads (Chemgenes, Cat # Macosko-2011-10) were size-selected using a 40 μm strainer, diluted to 350,000 per ml, and loaded onto 70 μm wide and 75 μm deep microfluidic device (Nanoshift). The nuclei and barcoded bead suspensions were loaded and run at 35 ml/hr each, along with carrier oil (BioRad Sciences, Cat # 186-4006) at 200 μl/min, to co-encapsulate single nuclei and beads in -75 μm drops (vol. -200 pl) at 4,500 drops/sec and double Poisson loading concentrations. The microfluidic emulsion was collected into 50 ml Falcon tubes for 10-25 min each and placed on ice 2 h before droplet disruption. Individual 200 μl reverse transcription (RT) reactions were performed on up to 90 K beads. After further exonuclease digestion, aliquots of 800-5 K beads were PCR amplified for 10 cycles, and PCR products were pooled in batches

of 4 wells or 16 wells for library construction. Purified cDNA was quantified, and 550 pg of each sample was fragmented, tagged, and amplified in each Nextera reaction. Libraries were sequenced on the Illumina NextSeq500 using between 1.6–1.7 pM and 0.3 μM Read1-CustSeqB (GCCTGTCCGCGGAAGCAGTGGTATCAACGCAGAGTAC) using a 20 × 8 × 60 read structure to a depth of 60,000 reads/nucleus.

### DroNc-seq read alignment and gene expression quantification

Raw sequencing reads were demultiplexed to FASTQ format files using bcl2fastq (Illumina; version 2.20.0). Digital expression matrices (DGE) were generated using the Drop-Seq tools pipeline (https://github.com/broadinstitute/Drop-seq, version 2.4.0) as follows. Cell and UMI barcodes were extracted from read 1 and tagged onto read 2 -- barcodes with any base quality score <10 were filtered out. Subsequently, reads were trimmed at the 5′ end to remove any TSO sequence and at the 3′ end to remove poly(A) tails and/or (reverse complemented) barcodes and adapters. Tagged and trimmed reads were aligned with STAR (version 2.7.3) against the GRCm38 genome assembly using the GEN-CODE M20 primary assembly genomic annotation, pre-filtered to remove pseudogenes. Gene counts were obtained on a per-barcode basis by summarizing the unique read alignments across exons and introns, collapsing UMI barcodes at hamming distance 1.

### Mouse dPnTg snRNA-seq data analysis

72 DGEs from DroNc-seq (42 from the PB-centered and 30 from Bar-centered dissections) sampling the dPnTg were imported into RStudio (R v 4.2.3) and converted into single Seurat objects; metadata were assigned to each object before merging them. An additional dataset sampling the entire Pons (that includes the dPnTg) was publicly available from the ABA effort to profile the whole mouse brain transcriptome at single-cell resolution using snRNA-seq (10X v3) and MERFISH techniques[25,26]. An AnnData file containing a single snRNA-seq DGE matrix and relative metadata representing the entire Pons was imported into RStudio and converted to a Seurat object using the *Convert()* and *LoadH5Seurat()* functions. Using the metadata annotation, which also includes the spatial localization from each snRNA-seq nuclei (imputed from MERFISH data), the Seurat object was subsetted to include only nuclei belonging to the dPnTg. Finally, the resulting object was merged with the DroNc-seq object. Nuclei with 1) mito-chondrial gene expression detection rate >10%; 2) hemoglobin gene expression detection rate >5%; 3) <400 or >10,000 unique gene features, possibly representing empty droplets/low-quality nuclei or cell doublets, respectively, were removed. A post-filtered dataset of 222,592 nuclei x 34,457 genes was inputted into Seurat v3.2.3 + Harmony v1.1 pipeline[27–30]. Downstream processing was performed using functionalities available in the Seurat R package. Data were first log-normalized using *NormalizeData()*, and then *CellCycleScoring()* was used to infer G2M and S cell cycle scores. This function classifies each cell into one of the 3 phases, G1, G2/M, and S, based on the expression of known G2/M and S phase marker genes[73]. Count data were then processed using *SCTransform()*, which performs a negative binomial-based normalization, identifies the top 3000 variable features, and regresses out covariates. Regressed covariates included sex, feeding schedule (fasted, re-fed, and ad libitum), CO2 treatment, mitochondrial gene detection rate, inferred cell cycle scores, experimental batch, library batch, mouse genotype, and anatomical dissection (PB-, Bar- and Pons-centered). Principal Component Analysis (PCA) was performed on the 3000 most variable features using the *runPCA()* function. *RunHarmony()* was subsequently used to harmonize the two technologies (snRNA-seq and DroNc-seq) gene expression profiles. Downstream analyses were conducted on the harmonized dataset. Distinct cell clusters were determined via Shared Nearest Neighbor (SNN) and k-Nearest Neighbor (KNN) analyses. For SNN analysis, resolution parameters of 0.4 for "all nuclei" and 0.6 for the neurons of the "excitatory" and "inhibitory" groups were used. T-distributed

stochastic neighbor embedding (t-SNE) was performed on the first 50 PCs to visualize cell clusters. Finally, DE analysis between clusters was performed using the non-parametric Wilcoxon Rank Sum test implemented in *FindAllMarkers()* and *FindMarkers()* functions. A gene was defined as differentially expressed if the absolute average log fold-change (avg_logFC) was >0.25 and the Bonferroni-adjusted *p*-value <0.01. Cell types were assigned to each cell cluster based on the expression of specific marker genes. Glia/non-neuronal cell types were removed. The remaining neuronal clusters were categorized into "excitatory" and "inhibitory" (see results). Expression datasets representing "excitatory" and "inhibitory" groups were re-processed the same way as described above. Descriptive statistics relative to the abovementioned datasets are presented in Supplementary Fig. 1a-i.

### Data analysis of the mouse PB scRNA-seq dataset from Pauli et al

scRNA-seq data were retrieved from the Pauli et al. manuscript that classifies the PB neuronal types by their transcriptional profile and axonal projections[38]. 4 DGEs representing 4 experimental batches were obtained from the NCBI Gene Expression Omnibus (GEO) portal (ID GSE207708) and imported into RStudio (R v4.2.3). A Seurat object, including only PB neurons, was generated and used for cluster analysis. DE between clusters was performed using the non-parametric Wilcoxon Rank Sum statistics implemented in *FindAllMarkers()* and *FindMarkers()* functions. A gene was defined differentially expressed if absolute logFC was >0.25 and Bonferroni-adjusted *p*-value < 0.01.

### MERFISH gene panel selection

MERFISH assay was performed by Vizgen, Inc. (Cambridge, MA, USA). (Vizgen, #10400003). A MERSCOPE panel of 315 genes meeting at least one of the following criteria was assembled: 1- highly variable genes obtained from DE analysis of the snRNA-seq dataset (adj. *p*-value <0.01; Av. logFC >0.25); 2- canonical glial, non-neuronal, and neuronal markers; 3- transcription factors, neuro-peptides, and receptors – including those which could be potential pharmacological targets. For each gene, a panel of 30 encoding probes was designed by Vizgen using a proprietary algorithm, except for 11 genes where the targetable regions were <30 (Supplementary Data 4). Each MERFISH encoding probe contains a tar-geting region complementary to the RNA of interest and a series of Vizgen's proprietary readout sequences that encode the specific barcode assigned to each RNA. In addition, 70 scrambled probes (blanks) to which have been assigned a specific binary barcode were added to the library as a negative control.

### MERFISH sample preparation

A total of 7 C57BL/6J mice (4 males and 3 females) 8-10 week-old from JAX were used for the MERFISH experiment. Mice were housed and sacrificed as described above. After decapitation, the brain was removed from the skull, chilled for 3 min in an ice-cold DMEM/F12, no phenol red (Thermo Fisher Scientific) media slush, and placed ventral surface up in an ice-cold stainless steel brain matrix (Roboz Surgical Instrument Co.). A 2 mm-thick coronal section containing the entire pons-medulla region was cut, placed in a square mold (S22, Kisker Biotech), embedded in OCT (Tissue-Tek® O.C.T. Compound, Sakura), and stored at −80 °C. Afterward, the brain block embedded in OCT was incubated for 1 h at −20 °C in a cryostat (LEICA CM1510 S CRYOSTAT), and 10 μm thick coronal sections were cut. To ensure the inclusion of our ROI, we cut from each mouse 10 sections at intervals of 80−90 μm starting approximately from −4.70 to −5.8 bregma level in the Franklin-Paxinos atlas[31]. Two sections at the time were mounted on a warm, functionalized, bead-coated MERSCOPE slide (Vizgen, #20400001) within the boundaries drawn using a 1cm² hexagonal gasket (Vizgen). Tissue sections were then placed face-up in a 60 mm petri dish (VWR, 25382-687) and stored at −20 °C. Subsequently, 4 ml of Fixation Buffer

(4% PFA; EMS, 15714) in buffered 1X PBS (ThermoFisher, AM9625) was added to each petri dish, and sections were incubated for 15 min at room T in a fume hood. After 15 min, the Fixation Solution was discarded, and the sections were washed 3 times, 5 min each, with a Washing Solution (1X PBS, ThermoFisher, AM9625) at room T. Then, 5 mL of 70% Ethanol (Sigma-Aldrich) was added to the petri dish, and sections were incubated for 5 min at room T. Finally, sections were transferred in a Polytube bag, 4 mm thickness (Vizgen) with 10 ml of 75% Ethanol (Sigma-Aldrich), sealed, and stored in the dark at 4 °C before shipping to Vizgen facility. After washing with 5 ml Sample Preparation Wash Buffer (Vizgen, #20300001) for 5 min and 5 ml Formamide Wash Buffer (Vizgen, #20300002) for 30 min at 37 °C, the sample was hybridized with the MERSCOPE Gene Panel Mix at 37 °C in an incubator for 36–48 h. The tissue slices were then washed twice with 5 ml Formamide Wash Buffer at 47 °C for 30 min and embedded into a hydrogel using the Gel Embedding Premix (Vizgen, #20300004), ammonium persulfate (Sigma, 09913-100G), and TEMED (N,N,N',N'-tetramethylethylenediamine) (Sigma, T7024-25ML) from the MERSCOPE Sample Prep Kit (Vizgen, #0400012). After the gel embedding solution polymerized, the sample was incubated with a Clearing Solution consisting of 50 μl of Protease K (NEB, P8107S) and 5 ml of Clearing Premix (Vizgen, #20300003) at 37 °C overnight. Then, the sample was washed with 5 ml Sample Preparation Wash Buffer and imaged on the MERSCOPE system (Vizgen 10000001). A fully detailed, step-by-step instruction on the MERFISH sample prep is available at: https://vizgen.com/resources/fresh-and-fixed-frozen-tissue-sample-preparation/. Full Instrumentation protocol is available at: https://vizgen.com/resources/merscope-instrument/.

### MERFISH imaging and cell segmentation

After image acquisition, the data were analyzed through the merlin pipeline through Vizgen's MERSCOPE Analysis Computer by selecting the watershed cell segmentation algorithm.

The output files for each coronal brain section consisted of (1) *cell_by_gene.csv*—A matrix where each row corresponds to a cell and each column to a gene. The matrix is not filtered for segmentation artifacts. Before analyses, cells with <15 gene counts were removed; (2) *detected_transcripts.csv*—DataFrame of all detected transcripts in a coronal section where each row is a detected transcript. The columns are "barcode_id"—315 internally used gene IDs that identify each gene univocally; "global_x, global_y"—the global micron x and y coordinates of each transcript; "global_z"—the index of the z-stack in the section where the transcript was detected. To note that 7 z-stacks per section were acquired at an interval of -1.5 μm; "x, y"—the pixel coordinates of a transcript within the field of view (FOV); "fov"—the index of the FOV where the transcript was detected; "gene"—the gene name of the detected transcript; (3) *cell_metadata.csv*—Spatial metadata of detected cells. Each row corresponds to a cell. The columns are: "fov"—the field of view containing the cell; "volume"—the volume of the cell in $\mu m^3$; "center_x"—the x coordinate of the center of the cell in global micron coordinates; "center_y"—the y coordinate of the center of the cell in global micron coordinates; "min_x, max_x"—the x minimum and maximum of the bounding box containing the cell in global micron coordinates; "min_y, max_y"—the y minimum and maximum of the bounding box containing the cell in global micron coordinates; (4) *cell_boundaries.hdf5*—Polygon boundaries relative to cells identified in a single FOV. Each file refers to a FOV. Boundaries are stored in.hdf5 format indexed by the unique cell ID; (5) *images*—Folder containing 7 mosaic_DAPI.tiff and 7 mosaic_PolyT.tiff images. These represent stitched DAPI or PolyT staining images acquired from a 10 μm thick MERFISH coronal section at -1.5 μm intervals; micron_to_mosaic_pixel_transform.csv—contains the transformation matrix used to convert micron into pixel coordinates; manifest.json—contains the metadata of the stacked image.

### Mouse dPnTg MERFISH data analysis

46 mosaic DAPI images, one per coronal section, were imported into Adobe Illustrator v26.5. Using the lasso tool, the dPnTg's boundaries were manually defined for each image. The cartesian pixel coordinates defining each image's boundaries were extracted using a custom script (Supplementary Data 34). Then, 46 gene count matrices (cell_by_gene.csv) related to the 46 DAPI images were imported into Python v3.8. Using the cartesian pixel coordinates defined by the lasso tool, the count matrices were subsetted to include only data relative to features (genes) and barcodes (cells) located within the defined boundaries. 46 subsetted matrices were imported into RStudio and converted into Seurat objects; metadata were assigned to each object before merging them[27,29,30]. Cells with <15 gene counts were filtered out. A post-filtered dataset of 685,289 cells x 315 genes was inputted into Seurat v3.2.3 + Harmony v1.1 pipeline[27–30]. Data were analyzed using the same bioinformatic pipeline employed for snRNA-seq with a few modifications. Briefly, count data were processed using *SCTransform()*. Regressed covariates included only mouse gender. PCA was performed on the 315 features using the *runPCA()* function. Harmony was subsequently used to harmonize the gene expression profiles across the sections. Downstream analyses were conducted on the harmonized dataset. Distinct cell clusters were determined via SNN and KNN analyses. SNN analysis was based on resolution parameters of 0.4 for "all cells", 0.8 and 0.6 for the neurons of the "excitatory" and "inhibitory" groups, respectively, 0.4 for the atlas 1 and 0.8 for atlases 2-3 (see results). T-SNE was used on the first 50 PCs to visualize cell clusters. Finally, DE analysis between clusters was performed using the non-parametric Wilcoxon Rank Sum statistics implemented in *FindAllMarkers()* and *FindMarkers()* functions. A gene was defined as differentially expressed if the absolute average log fold-change (avg_logFC) was >0.25 and the Bonferroni-adjusted $p$-value <0.01. As in snRNA-seq analysis, after assigning all the clusters to a cell type, clusters corresponding to glial/non-neuronal cell types were discarded. The remaining neuronal clusters were divided into "excitatory" and "inhibitory". They underwent the same analyses as described above. Descriptive statistics relative to the abovementioned datasets are in Supplementary Fig. 6a–h. Next, raw and normalized gene count matrices, metadata, and cartesian pixel coordinates of each polygon were extracted from the three Seurat objects containing "all cells", "excitatory", and "inhibitory" neurons and imported into GIOTTO v1.1.2 package for data visualization[74]. The function *createGiottoObject()* was used to create a single GIOTTO object, which included dPnTg cells and transcripts across 46 sections. *subsetGiottoLocs()* was employed to subset the gene count matrices based on spatial coordinates to generate the 4 anatomical subregions that were then analyzed using the Seurat v3.2.3 + Harmony v1.1 pipeline described above.

### Estimation of clusters' replicability using MetaNeighbor

The R package MetaNeighbor v1.14.0[39,40] was employed to assess cluster replicability across technologies (i.e., MERFISH, snRNA-seq(10X), DroNc-seq, scRNA-seq (10X)) and species (i.e., *Homo Sapiens*, *Mus Musculus*). Four main comparisons were made using MetaNeighbor: (1) across technologies, between MERFISH and snRNA-seq neuronal datasets of the mouse dPnTg and (2) between MERFISH and scRNA-seq neuronal datasets[38] of the mouse PB; (3) across species, between the mouse and the human[68] snRNA-seq neuronal datasets of the dPnTg and (4) between the mouse scRNA-seq[38] and human snRNA-seq[68] of the PB. For the cross-species analyses (points 3-4), gene symbols were converted between species using a manifest file ("gene_orthologs.gz") listing gene symbol correspondences across species as available at NCBI (https://ftp.ncbi.nlm.nih.gov/gene/DATA/).

Briefly, unique IDs were assigned to neuronal clusters of the two datasets. Seurat objects were converted into SingleCellExperiment objects using the function *as.SingleCellExperiment()*. The two objects

were then merged using the *mergeSCE()* function from the Meta-Neighbor package. The function selects only genes, assays, and metadata columns shared by the two objects. The function *variableGenes()* was used to select genes with high variance in both datasets. In the comparison between MERFISH and snRNA-seq/ scRNAseq (points 1–2), the 315-panel genes were set as highly variable genes. The unsupervised *MetaNeighborUS()* function with the "fast_version" parameter set to TRUE and the "symmetric_output" parameter set to FALSE was used to assess cell type homology. In brief, cells from the reference dataset (e.g., MERFISH) vote for their closest neighbors in the target dataset (e.g., snRNA-seq), effectively ranking these cells by similarity. Then, the cell-level ranking is aggregated at the cell-type level (i.e., clusters) in the target dataset as an area under the receiver operator characteristic curve (AUROC), which mirrors the proximity of a target cell type to the reference cell type. The same analysis is computed by reversing reference and target roles. The *topHitsByStudy()* function was used to select only matches with an AUROC >0.85 and/or classified as "reciprocal" top hits.

## Functional classification of gene sets driving cell type replicability

In cross-species analysis (point 3 in the above paragraph), we conducted gene ontology (GO) enrichment analysis of the gene sets driving the cluster replicability. A list of GO sets (*Mus Musculus*) comprising 22,546 GO terms categorized into the three main classes, Cellular Component (CC), Molecular Function (MF), and Biological Process (BP), was downloaded from https://figshare.com/articles/dataset/Protocol_data_R_version_/13020569/2[40]. The GO sets were filtered to (i) include only genes shared with our merged human-mouse dataset and (ii) be large enough to learn expression profiles (>10 genes) and small enough to enrich for GO terms (<100), as previously described[40]. Finally, the supervised *MetaNeighbor()* function was employed to construct a rank correlation network between cells for a GO gene set and predict cell type membership. The resulting AUROC, in this case, represents how well cells can be assigned to a cell type label using individual GO gene sets (how well a gene set contributes to each cell-type replicability). AUROC values of ~0.5–0.6 indicate random performance, AUROC values of ~0.7 suggest that they contribute moderately to replicability, while AUROC values >0.8 indicate high performance[40].

## Estimation of clusters' replicability between snRNA-seq and the 4 MERFISH subregion atlases using CCA for spatial dashboard

To compare mouse neuronal clusters resolved by snRNA-seq versus those resolved by MERFISH, we applied a canonical correlation analysis (CCA) function built in Seurat v3.2.3 that operates at the single-cell level, and then we aggregated the results at the cluster level to calculate the cluster-to-cluster correspondence. The following functions built-in Seurat v3.2.3 were used: (1) *FindTransferAnchors()*, which performs a CCA on the reference (snRNA-seq) and query (MERFISH) and identify cell anchors which are used to transfer data from the reference to the query; (2) *TransferData()* to transfer labels across single-cell datasets. The function's output includes a *prediction score* for each MERFISH cell mapping onto each snRNA-seq cluster and a *max prediction score* with the respective *predicted ID*, i.e., the predicted snRNA-seq cluster for each MERFISH cell ID with the highest *prediction score*. We aggregated the results at the cluster level by considering only those matches with a number of cells mapped from snRNA-seq clusters onto MERFSH clusters and vice versa >0.25%, after normalizing for the cluster size.

## Specificity, sensitivity, and reproducibility of MERFISH assay

MERFISH efficiency was evaluated by measuring the number of transcripts per FOV (FOV size = 200 ×200 μm). Only slices with >25,000 transcript counts per FOV were retained (Supplementary Fig. 3a). As a

control for MERFISH specificity, for all the sections was demonstrated (1) Pearson's *r* correlation coefficient >70% with a bulk RNA-seq dataset from the whole mouse brain (Supplementary Fig. 2f, 3b, Supplementary Data 35-36) and (2) a difference of 15.5 folds from the non-specific signal (Supplementary Fig. 5a). For the MERFISH dataset, the average expression of the 315 genes was calculated across all cells and is reported as log10 raw counts (or log10 (raw counts+1)). For the bulk RNA-seq dataset, the average expression of the 315 genes was calculated across all samples and is reported as log10 FPKM (Fragments per kilobase of transcript per million mapped fragments). Pearson's *r* correlation between the average expression values of the 315 genes in MERFISH and bulk RNA-seq datasets was performed by matching the same isoform between the two sources (codebook Supplementary Data 35-36). The bulk RNA-seq dataset from the whole mouse brain can be retrieved at https://www.ebi.ac.uk/arrayexpress/experiments/E-MTAB-6081/.

Experimental reproducibility was evaluated by computing the Pearson's *r* correlation coefficient of the average gene expression of 315 genes between sections of the same mouse (intra-batch reproducibility) and sections of different mice (inter-batch reproducibility) (Supplementary Fig. 4a, b). Sequential sections exhibited a higher pairwise correlation compared to non-sequential sections. (Supplementary Fig. 4a, b, 5c) In addition, the correlations between two coronal sections from the same or two different mice, representing approximately the same bregma level, were always extremely high ($r > 0.99$, $p = 0$) (Supplementary Fig. 5d, e). The difference in sensitivity between MERFISH and snRNA-seq was estimated by computing the fold change between the average expression levels of 315 genes across all cells in snRNA-seq versus MERFISH datasets. Average gene expression was 0.5 folds higher in MERFISH compared to the snRNA-seq dataset (Supplementary Fig. 5b), indicating a nearly identical sensitivity.

## Interactive visualization of MERFISH and snRNA-seq data

The design and realization of a dashboard able to produce interactive visualization of spatial-transcriptomic data were done in partnership with HEAVY.AI. The dashboard hosts two viewers on two different pages: the first viewer, called "spatial cell viewer," displays a total of 14 full, 10 μm thick coronal sections and covers at an interval of 80–90 μm a region from −4.7 to −5.8 bregma level in the Franklin-Paxinos atlas[31], whereas the second viewer, called "subregion cell viewer", hosts the data relative to the four subsetted regions (see section " MERFISH data analysis"). The dashboard can be accessed at: http://harvard.heavy.ai:6273/. Complete documentation can be found at: https://docs.heavy.ai/?_ga=2.207206352.2137306788.1595867219-1426127794.1594677732. Transcriptomic data for the 3 snRNA-seq and 7 MERFISH datasets, all the raw and normalized count matrices, the cell metadata, and the t-SNE embeddings were uploaded on the single-cell BROAD portal. The study can be accessed at: https://singlecell.broadinstitute.org/single_cell/study/SCP1808. Complete documentation can be found at: https://singlecell.zendesk.com/hc/en-us.

## Human dPnTg snRNA-seq dataset data analysis

snRNA-seq data were retrieved from a recent publication by Siletti et al., profiling the whole transcriptome of the entire adult human brain at a single-nucleus resolution[68]. A single .loom file containing a prefiltered DGE (for low-quality nuclei/doublets) of over 3 million nuclei and relative metadata was imported into RStudio. *As.Seurat()* was employed to convert the .loom file into a Seurat object. The Seurat object was then subsetted to include only nuclei from anatomical dissections of (1) the pontine reticular formation (PnRF) and the PB or (2) other nuclei in the dPnTg and the DTg. A subsetted object of 50,250 high-quality nuclei x 37,165 genes was imported into RStudio (R v4.2.3) and processed using the Seurat v3.2.3 + Harmony v1.1 pipeline[36–39]. Data were analyzed using the same bioinformatic pipeline employed

for snRNA-seq and MERFISH with a few modifications. Briefly, count data were processed using *SCTransform()*. Regressed covariates included age, cell cycle score, 10X chemistry, mitochondrial gene detection rate, donor label, and anatomical dissection. The data were derived from 3 male donors. PCA was performed on the 3,000 most variable features. *RunHarmony()* was subsequently used to harmonize the gene expression across different donors. Distinct cell clusters were determined via SNN and KNN analyses in Seurat. SNN analysis was based on resolution parameters of 0.4 for "all nuclei", 0.8 for the neurons of the "excitatory", "inhibitory" groups, and for all PB neurons. T-SNE was used on the first 50 PCs to visualize cell clusters. DE analysis between clusters was performed using the non-parametric Wilcoxon Rank Sum statistics implemented in *FindAllMarkers()* and *FindMarkers()* functions. A gene was defined as differentially expressed if the absolute average log fold-change (avg_logFC) was >0.25 and the Bonferroni-adjusted *p*-value <0.01. Consistently with MERFISH and snRNA-seq analyses, all the clusters were assigned to a cell type. Clusters corresponding to glia/non-neuronal cell types were discarded. The remaining neuronal clusters were divided into "excitatory" and "inhibitory". They underwent the same processing as described above.

### RNA scope in situ hybridization
RNA Scope Multiplex Fluorescent Reagent Kit V2 (Advanced Cell Diagnostics, Hayward, CA; Cat. #323100) was used to perform RNA scope in situ hybridization for *Pdyn*, *Gpr101*, and *Foxp2* mRNA. Mice were, first, intracardially perfused with formalin (10% buffered solution) under deep anesthesia induced by isoflurane exposure (5% in O2), and then brains were removed from the skull and post-fixed in formalin (10% buffered solution) overnight. After incubating in 20% sucrose (for cryoprotection) for 24 h, the brains were cut into 30 μm-thick sections. Sections were treated with protease (40 ˚C; 30 min; Protease IV, RNA scope) and incubated with RNA scope probes for *Pdyn*-C1 (RNA scope® Probe- Mm-Pdyn; Cat. #318771), *Gpr101*-C2 (RNA scope® Probe- Mm-*Gpr101*; Cat. #317281), and *Foxp2*-C3 (RNA scope® Probe-Mm-Foxp2; Cat. #428791; Advanced Cell Diagnostics) during the hybridization step (2 h; 40 °C). After the hybridization step, we performed three amplification steps (40 °C; AMP1-FL and AMP2-FL: 30 min each; AMP3-FL: 15 min), followed by horse radish peroxidase-C1 (HRP-C1) amplification (40 °C; 15 min). Sections were then incubated in TSA plus Fluorescein (Perkin Elmer, Cat. #NEL744001KT) to visualize *Pdyn* mRNA (Channel 1 at 488 nm) in green. This is followed by incubating the sections in HRP-C2 amplification step (40 °C; 15 min). Sections were then incubated in TSA plus Cy3 (Perkin Elmer, Cat. #: NEL754001KT) fluorophore (1:1000; 30 min) to visualize *Gpr101* mRNA (Channel 2 at 550 nm) in red. In the last step of the process, sections were subjected to HRP-C3 amplification (40 °C; 15 min) followed by TSA plus Cy5 incubation (40 °C; 30 min; Perkin Elmer; Cat. #NEL754001KT) to visualize *Foxp2* mRNA (Channel 3 at 647 nm) in magenta. After each fluorophore step, sections were subjected to HRP blocking (40 °C; 15 min). After each step in the protocol, the sections were washed two times with 1X wash buffer provided in the kit. The covered sections (Vectashield mounting medium; Vector Laboratories) were imaged and photographed with a confocal microscope (Leica Stellaris 5) at final magnification of 20X and 63X.

### Graphics
All graphic representations were generated using R (v. 4.2.3) base functions or R packages. Bar plots, scatter plots, box plots, donut plots, staked area charts, line charts, and correlation matrix heatmaps were generated with R base functions or the ggplot2 package[75]. Sankey plots were generated with the networkD3 package. Dot plots, t-SNEs, and violin plots were generated using functions built in the Seurat v3.2.3 package[30]. Voronoi plots were generated using the functions built in the GIOTTO v1.1.2 package[74]. Human-mouse dot plots were generated using the function built in the MetaNeighbor v1.14.0

package[39,40]. The schematic in Fig. 1a was created with BioRender.com. The schematic in Fig. 2a was created using Adobe Illustrator. The schematics in Figs. 7e and h were created using Inscape.

### Statistics and reproducibility
No statistical method was used to predetermine the sample size. Our sample sizes for MERFISH and snRNA-seq are similar to or larger than those reported in the literature[24,76–78]. No randomization or blinding was performed for sample collection and data analysis. This was not required since we did not perform any comparison between different conditions or treatments. The criteria used to exclude data during the quality control process for mouse MERFISH, mouse snRNA-seq, and human snRNA-seq are documented in the "Mouse dPnTg MERFISH data analysis", "Mouse dPnTg snRNA-seq data analysis", and "Human dPnTg snRNA-seq data analysis" sections, respectively.

### Reporting summary
Further information on research design is available in the Nature Portfolio Reporting Summary linked to this article.

## Data availability
The mouse DroNc-seq raw and processed data generated in this study have been deposited in the GEO database under accession code GSE226809. The mouse snRNA-seq raw and processed data from Allen Brain Institute[25,26] used in this study are available in the Allen Brain Atlas database https://knowledge.brain-map.org/data/LVDBJAW8BI5YSS1QUBG/collections. The mouse MERFISH raw and processed data generated in this study have been deposited in the Beth Israel Deaconess Medical Center database: https://research.bidmc.harvard.edu/datashare/DataShareInfo.ASP?Submit=Display&ID=7. The mouse PB scRNA-seq raw and processed data from Pauli et al. [38]. used in this study are available in the GEO database under accession code GSE207708. The human snRNA-seq raw and processed data from Siletti et al. [68]. used in this study are available in the Google bucket: https://storage.cloud.google.com/linnarsson-lab-human. Source data are provided with this paper.

## Code availability
MERlin pipeline used to process MERFISH raw data is available on Zenodo: 10.5281/zenodo.3758540. R and Python code used to generate results in the manuscript are available on Zenodo: https://zenodo.org/records/10103722? and 10.5281/zenodo.10396868.

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

## Acknowledgements

The authors thank Dr. Jiang He and Dr. George Emanuel from Vizgen for consenting to us enrolling in the Vizgen early access program and granting us access to MERSCOPE before commercialization. We thank Mr. Tony Duarte from HEAVY.AI for his availability and collaboration in implementing our vision to use their HEAVY.AI proprietary GPU-accelerated technology to query single-cell spatial transcriptomic datasets in real-time. This project was funded by the NIH grants R01DK075632 (B.B.L.), P01HL149630, NS072337 (C.B.S.), R01-DK125708 (A.M.J.V.), R01-DK113030, P20-DK119789 (M.L.Z.), R01-NS091126 (E.A.), R00HL144923 (J.M.R.), DP1-AT010971, R01-MH12343 (M.L.A.).

## Author contributions

Project design: B.B.L., S.N. A.M.J.V. Project management: B.B.L., S.N., A.M.J.V., R.D.L. Data generation (snRNA-seq): S.N., A.M.J.V., N.K., R.D.L., J.M.R., D.P., M.V. Data generation (MERFISH): S.N., R.D.L., N.K. Data processing (snRNA-seq): C.L.J, H.S. Data processing (MERFISH): S.N., Data analysis (snRNA-seq): S.N., A.Z. Data analysis (MERFISH): S.N., A.Z. Data storage (snRNA-seq/ MERFISH): O.A., S.N. Spatial dashboard design: S.N., D.N., C.B. Animal experiments: A.M.J.V., N.K., Z.Y., R.M.G, S.N, R.D.L. In situ hybridization: R.D.L. Anatomic cluster annotation: S.N., R.D.L., C.B.S., A.M.J.V., N.K. Drafting of the manuscript and figure generation: S.N., R.D.L., E.A, A.M.J.V., N.K. Manuscript review and editing: S.N., B.B.L, A.M.J.V., C.B.S., A.Z., R.D.L., E.A., M.L.A., L.T., K.D.H., M.L.Z. Vizgen team contributed to MERFISH data generation and data processing. HEAVY.AI team contributed to the spatial design and implementation of the spatial viewer. B.B.L., C.B.S., and A.M.J.V. contributed equally to this work.

## Competing interests

Cory Brannigan is a software architect at HEAVY.AI. The remaining authors declare no competing interests.

## Additional information

[1]Department of Medicine, Division of Endocrinology, Diabetes and Metabolism, Beth Israel Deaconess Medical Center and Harvard Medical School, Boston, MA, USA. [2]Broad Institute of MIT and Harvard, Cambridge, MA, USA. [3]Department of Neurology, Division of Sleep Medicine, Beth Israel Deaconess Medical Center and Harvard Medical School, Boston, MA 02215, USA. [4]Department of Twin Research & Genetic Epidemiology, King's College London, London, UK. [5]Division of Nephrology, Department of Medicine, Beth Israel Deaconess Medical Center, Harvard Medical School, 330 Brookline Ave, Boston, MA 02215, USA. [6]UCL Queen Square Institute of Neurology, University College London, London, UK. [7]HEAVY.AI, 100 Montgomery St Fl 5, San Francisco, California 94104, USA. [8]Department of Neuroscience and Pharmacology, University of Iowa, Iowa City, IA, USA. [9]Fraternal Order of Eagles Diabetes Research Center. University of Iowa Carver College of Medicine, Iowa City, IA 52242, USA. [10]Present address: Department of Molecular Biology, Massachusetts General Hospital, Boston, MA, USA. [11]Present address: Department of Genetics, The Blavatnik Institute, Harvard Medical School, Boston, MA, USA. [12]These authors contributed equally: Roberto De Luca, Antonino Zito, Nataliya Klymko ✉e-mail: aversteg@bidmc.harvard.edu; csaper@bidmc.harvard.edu; blowell@bidmc.harvard.edu

