## [Peer Review File · Nature Communications]

A spatially-resolved transcriptional atlas of the murine dorsal pons at single-cell resolutionREVIEWER COMMENTS

Reviewer #1 (Remarks to the Author):

In this manuscript, the authors performed the single-nucleus RNA-seq (snRNA-seq) and generated a transcriptomic atlas of "dorsal pontine tegmentum" (dPnTg) of mouse. Further, the authors employed MERFISH to map the distribution of 315 informative genes hints from snRNA-seq to map the subtypes spatially. They compared the neuronal subtypes of dPnTg between mouse and humans and found many subtypes were transcriptionally similar. Finally, the authors developed a freely-accessible website to provide access to this spatially-resolved transcriptional dataset. This study represents a valuable resource for further study of the function of these nuclei. This reviewer has several concerns.

1. The authors profiled 447,833 nuclei, but only 149,159 nuclei finally passed the quality control steps (line 109, page 3). Why did two thirds of nuclei unqualified? Does this mean the quality of snRNA-seq is low?

2. To reveal the correspondence between MERFISH and DroNc-seq clusters of the dPnT, the authors applied an unsupervised replication framework and depicted the clusters correspondence between MERFISH and DroNc-seq datasets (Fig 2M). However, these cluster-to-cluster correspondence lost the single cell resolution. Is it possible to examine the correspondence with single cell resolution? Besides, the authors mentioned "the correspondence between MERFISH and DroNc-seq clusters was reciprocal in 38/84 instances (line 211, page 5)". The ratio of reciprocal correspondence is less than 0.5. Can the author speculate the possible reason?

3. The spatial distribution pattern of MERFISH clusters is not easy to discriminate due to the mixture of clusters in the figure (Fig 3F, 4A, 4F, 5A, 6A). The authors can present a voronoi plot showcase of important cluster. Besides, the authors can add the cell density curve of MERFISH cluster along medial-lateral and dorsal-ventral axis.

4. Previous study has showed the spatial transcriptomics of PBN in mouse (Fu et al., 2022, Cell). The authors should compare their MERFISH data with those as well.

5. The author compared the transcriptomic subtypes between mouse and human with snRNA-seq data and found that "33/52 human snRNA-seq clusters corresponded to 28/62 mouse DroNc-seq clusters, and the correspondence was mutual in 23/38 instances (Fig 8H-I; STable 30) (line 601, page 12)." It is not clear how mutual correspondence was calculated. How could the ratio of mutual correspondence be higher than the ratio of unidirectional correspondence?

6. The authors mentioned "All neurons in this group expressed Slc32a1 (Vgat)." (line 142, page 3). However, the dot plot (Fig 1L) didn't support this conclusion because ratio of Slc32a1+ neurons in these named "inhibitory neurons groups" is much smaller than 1.

Reviewer #2 (Remarks to the Author):

In this manuscript the authors make use of two powerful technologies, MERFISH and single-nucleus (sn) transcriptional profiling to obtain a very powerful and highly resolved map of the dorsal pontine tegmentum (dPnTg) of the murine brain. They subsequently use established methods and approaches to cluster and subcluster the different cell types within the dPnTg and performed pairwise comparisons for cluster replicability across both technologies and with data from human snRNA-seq samples. Of note, to visualize and make this large dataset more accessible the authors also invested time in creating an interactive dashboard which contains selection tools, zoom functions and several informative plots that display metadata. Overall, the open-access repository of all the raw data and of

the visualized pre-processed data serves as an invaluable resource, empowering researchers in this field and has a great potential to facilitate rapid unraveling of novel functions and biological processes within this intricate brain tissue.

Overall, I applaud the authors for making such a detailed and unprecedented dataset and for their diligence in disentangling all the different cell types within the dPnTg. I do have a few major and minor comments that I hope can further improve the readability and strength of the manuscript.

Major comments:

My main issue concerns readability of the paper. Each figure contains a lot of panels and information and it's extremely difficult to read the details. I believe it would be beneficial to reduce the number of panels in the main figures and move supporting figures to supplemental data. Similarly, it would help if the authors could provide more schematics for some of the figures that would depict their experimental or data analysis strategy. For example, in figure 2 they display a simplistic MERFISH pipeline but it would help more if they could explain how many mice and serial sections (80-90 μm intervals) they took from each. Is this in fact a (serial) 3D dataset from the same mouse or an aggregation of different mice.

Minor comments:

1. Most of the analysis strategy focuses on cell type identification (and cross data comparison) and there is a missed opportunity to explore the presence of spatial patterns (genes, niches, regions, domains, ...). Can the serial sections be considered a 3D dataset and be used to identify brain structures and expression in 3 dimensions? Besides visualizations of cell type distribution changes within the sections there hasn't been much done. Is this the consequence of issues with registering the serial images?

2. The GPU-accelerated visualization dashboard is potentially a powerful tool for researchers to display similar datasets in the future, however it is unclear whether this dashboard is specifically tailored for this dataset or could also be used with external datasets.

REVIEWERS COMMENTS

Overall, we thank the reviewers and editor for their very constructive comments. This led to a number of changes and improvements, which we believe have greatly enhanced the quality of our study. A bullet list of all our revision's major changes/ improvements is summarized immediately below. After that are our detailed responses to each of the reviewers' comments.

Summary of major changes/ improvements to the manuscript.

1. Integration of our DroNc-seq data with Allen Brain Atlas snRNA-seq data (10X.v3) from the same region of interest, the mouse dPnTg, has significantly improved the statistical power, snRNA-seq metrics, clustering granularity, and correlation with the MERFISH dataset.
2. Transcriptional correspondence of neuronal clusters from MERFISH atlases 1-4 with snRNA-seq neuronal clusters of the mouse dPnTg has been assessed by MetaNeighbor analysis. Results have been replicated by canonical correlation analysis (CCA) computed at the single-cell level.
3. Cell clustering and related analyses on the human dPnTg snRNA-seq dataset have been re-performed after integrating additional data made available in the Siletti et al. manuscript. MetaNeighbor-inferred correspondences between mouse and human snRNA-seq clusters of the dPnTg have also been recomputed.
4. The manuscript has been drastically shortened from over 8,000 to <5,500 words and modified according to the Nature Communication guidelines and the reviewer's comments.
5. Figures and tables have been simplified by reducing the number of panels while keeping the most relevant insights. Non-central findings and all technical analyses have been moved into the supplementary methods.
6. The figure schematics for the snRNAseq and MERFISH studies have been redesigned to give a quick overview of the experimental flowchart. Implementing points n. 4-6 has significantly improved the manuscript's readability.
7. BROAD single-cell and HEAVY.AI spatial dashboards plotting mouse snRNA-seq and MERFISH data have been updated according to the new analysis to allow the reviewers to access the data easily.
8. Finally, we have made available raw/ preprocessed data and the code used for the analyses, and we have tested and confirmed the reproducibility of the results.

Reviewer #1 (Remarks to the Author):

In this manuscript, the authors performed single-nucleus RNA-seq (snRNA-seq) and generated a transcriptomic atlas of the "dorsal pontine tegmentum" (dPnTg) of the mouse. Further, the authors employed MERFISH to map the distribution of 315 informative genes hints from snRNA-seq to map the subtypes spatially. They compared the neuronal subtypes of dPnTg between mice and humans and found that many subtypes were transcriptionally similar. Finally, the authors developed a freely-accessible website to provide access to this spatially-resolved transcriptional dataset. This study represents a valuable resource for further study of the function of these nuclei. This reviewer has several concerns.

1. The authors profiled 447,833 nuclei, but only 149,159 nuclei finally passed the quality control steps (line 109, page 3). Why were two-thirds of nuclei unqualified? Does this mean the quality of snRNA-seq is low?

The reviewer raised an important point. We provide our answer to this question in the following points below.

1. Generally, single-nuclei RNA-seq detects fewer genes per nucleus than single-cell RNA-seq, as only the nuclear fraction of the transcriptome is sequenced. In fact, the number of transcripts in the nucleus represents ~30% of the total mRNA fraction, depending on the cell type (PMID: 30586455). Consequently, it is more likely that more nuclei than whole cells are discarded in QC steps that involve filtering out nuclei with a low gene detection rate. We also highlight that the number of detected genes per nucleus set as a cutoff to exclude potential empty droplets/ low-quality nuclei from downstream analyses was in line with the commonly used cutoff values.

Notably, despite 2/3 of the nuclei being discarded, the number of nuclei used (N=149,159) for downstream analyses was still higher than most of the other published single-cell RNAseq atlases (Fig-1).

2. In addition, profiling nuclei transcriptomes offers more advantages than whole cells in neuroscience research. For instance, it avoids alterations in transcription, such as the expression of early-immediate genes (IEGs), that could distort the actual cellular transcriptional profile; it also avoids the possibility of some of the larger cells being excluded during the filtering process and/ or failing to survive the dissociation steps, which prevents bias in cell type representation. Nowadays, most sequencing studies in neuroscience are performed on nuclei.

3. Our DroNc-seq assay was performed as the first step of this study using an in-house protocol. Noteworthy, we used the same setup adopted in the Aviv Regev Lab (PMID: 28846088). Our QC'ed dataset shows a gene detection rate highly consistent with the abovementioned paper (mean~1,300 genes/nucleus). While we are aware that in these past 5 years, the single-nuclei protocols have greatly improved, leading to better assay sensitivity and efficiency, we would also highlight that our QC'ed dataset (N=149,159 nuclei) provided sufficient granularity and statistical power to transcriptionally characterize, in detail, our region of interest. We also reasoned that increasing the stringency of our cutoff could be beneficial despite further reducing the number of nuclei available for analyses. To this end, we run cell Bender (PMID: 37550580) on the DroNc-seq data to estimate a threshold of a number of genes per nucleus capable of including only true positive nuclei. Cell Bender suggested discarding nuclei with <400 detected genes, which resulted in a dataset of ~95,000 nuclei. We used this dataset as input to an integration procedure outlined in point 4 below.

4. Finally, for this revision, we put effort into increasing the sample size and further improving the snRNA-seq metrics of the study by integrating our DroNc-seq (~95,000 dPnTg nuclei) data with a more recent snRNA-seq (10X_v3) (~125,000 dPnTg nuclei) dataset from the ongoing Allen Brain Atlas (ABA) study sampling the same regions of the pons [doi: <https://doi.org/10.1101/2023.03.06.531121>; doi: <https://doi.org/10.1101/2023.03.06.531348>]. DroNc-seq data integration with the ABA snRNA-seq dataset and the use of a more stringent cutoff for downstream analyses, as described in point 3, improved the sample size and statistical power of the study, the snRNA-seq metrics, clustering granularity, the ability to detect low-expressed genes/ genes with low turnover and the cluster-to-cluster correspondence with MERFISH dataset (MetaNeighbor analysis). More importantly, in our revised dataset, the number of nuclei used for downstream analyses is higher than those from published single-cell RNAseq atlases that investigated brain regions of similar size, and the median number of genes per cell aligns with recent field works (Fig-1).
5. Accordingly, in the revised manuscript, we have rearranged the result section related to the mouse dPnTg cell type identification using snRNA-seq (lines 92- 128) and updated all relative panels in Fig 1 (except panel 1b), Supp. Fig. 1-2 and Supp. Tables1-3. Likewise, results from MetaNeighbor analysis performed between snRNA-seq neuronal clusters of the mouse dPnTg versus the cluster annotation of the 4 MERFISH atlases (Fig. 8a-b; Supp. Table 26) and versus the cluster annotation of the human dPnTg (Supp Fig 12-14; Supp. Tables 30-31) have been updated as well and moved to line 411-- 424 and line 441– 468, respectively.

2. To reveal the correspondence between MERFISH and DroNc-seq clusters of the dPnTg, the authors applied an unsupervised replication framework and depicted the clusters correspondence between MERFISH and DroNc-seq datasets (Fig 2M). However, this cluster-to-cluster correspondence lost the single-cell resolution. Is it possible to examine the correspondence with single-cell resolution? Besides, the authors mentioned, “the correspondence between MERFISH and DroNc-seq clusters was “mutual” in 38/84 instances (line 211, page 5)”. The ratio of “mutual” correspondence is less than 0.5. Can the author speculate on the possible reason?

The reviewer raised two constructive points that we have addressed below as follows:

1. **“loss of single-cell resolution when applying**

MetaNeighbor to assess cluster-to-cluster correspondence between MERFISH and snRNA-seq datasets". MetaNeighbor is optimized to give back robust statistics, which operates better at the cluster level. Theoretically, it would be possible to run it on individual cells, but there would be no way of scoring the goodness of the integration because the statistic, in this case, relies on how well the cells are grouped. In addition, because MetaNeighbor does not use any threshold, a cell would not be automatically placed into one category instead of another. We deem that a correspondence computed at the cluster level is correct and robust enough to impute transcriptional information between the two datasets. However, for visualization purposes, such as in our HEAVY.AI dashboard, where data are displayed at the single-cell level, we performed a canonical correlation analysis (CCA) at the single-cell level. We then aggregated the single-cell results at the cluster level to calculate the cluster-to-cluster correspondence and compare the MetaNeighbor output with the CCA output (see methods for a detailed explanation of thresholds). The two methods agreed in 73/105 (70%) instances. (Fig-2). Setting a more stringent threshold would significantly increase the agreement between the two methods, even though the number of investigated matches would drop.

2. **"The ratio of "mutual" correspondence is <0.5"**.

This point requires a detailed explanation. In MetaNeighbor, we set "symmetric_output= FALSE", meaning each cluster is tested against only two clusters in the target dataset (closest and second-closest match). This representation helps to rapidly identify a cluster's closest hits as well as its closest outgroup. The nonsymmetric view makes it clear when best hits are not reciprocal. The lack of reciprocity in voting is an important tool for detecting imbalances in dataset composition. In our new MetaNeighbor analysis

(line 410-423; Fig 8 manuscript) performed between the neuronal clusters of the 4 MERFISH atlases and snRNAseq (10X + DroNc-seq) neuronal clusters, we have a "mutual" match in 50/122 instances (40%). This result might depend on several factors: **i- the AUROC threshold set.** In Fig-3, we show the distribution of all the matches with an AUROC > 0.85 that are classified as "mutual" or "non-mutual" hits. "Non-mutual" hits have a lower median and show a bimodal distribution. Therefore, raising the threshold to include matches >0.9 (instead of >0.85) and/or "mutual" hits will drastically improve the % of "mutual" hits over the total hits, bringing it to >55%. However, this will cause the exclusion of some biologically relevant, "non-mutual" correspondences; **ii- differences in the region of interest (ROI) sampled in snRNA-seq (10X.v3 + DroNc-seq) versus MERFISH atlases 1-4.** Specifically, stereotactic dissections of the dPnTg done by this study and the ABA effort included unavoidably marginal regions external to our ROI that could not be excluded *a priori* since single cells do not hold any spatial information. On the contrary, MERFISH clusters from atlases 1-4 mapped only to our ROI, the dPnTg, since it was possible to filter out clusters from neighbor regions, given the technology holds spatial information. Clusters that, for this reason, will not have their corresponding match in the comparing dataset will map onto the closest transcriptional cluster, but the match's AUROC score will be lower (since they are not transcriptionally identical), and the match will be "non-mutual". Most of these matches will not pass the AUROC threshold set, though.

iii- the difference in cluster granularity between the two approaches leads to fewer "mutual" matches. The difference in the number of variable features used to cluster the two datasets (315 vs. 3000) led to snRNA-seq resolving more (>30%) neuron types

compared to MERFISH when we clustered the neurons by dividing them into “excitatory” and “inhibitory” groups (Figs 1-2 manuscript). To partially mitigate the difference in cluster granularity between the two technologies, we compared the snRNA-seq neuronal clustering with the clusters from the 4 non-overlapping MERFISH subregion atlases that achieved a better granularity because clustering was done on smaller regions than the whole PnTg (lines 410—423; Fig 8; STable 26 manuscript). This led to identifying more “mutual” matches and gave a more resolved representation of neuronal types' transcriptional and spatial organization in the dPnTg. Regarding “non-mutual” matches, either a MERFISH cluster corresponded to more than one snRNA-seq cluster or viceversa. In both scenarios, one technology could resolve one or more neuronal types better than the other. In the first scenario, snRNA-seq led to better granularity than MERFISH; a possibility would be that more snRNAseq clusters matching with only one MERFISH cluster represent subclusters of the main MERFISH population that could not be resolved into subpopulations for the absence of one or more marker genes. The opposite scenario happens because clustering smaller regions, as we did for the 4 MERFISH atlases, also leads to better cluster granularity than clustering the whole dPnTg. **iv) transcriptionally identical neuron types in the 4 MERFISH atlases lead to “non-mutual” matches with snRNAseq.** A clear example is the Sst neurons, which in the snRNAseq dataset cluster all together (cluster_ex_33), whereas in the MERFISH dataset, composed of 4 atlases clustered independently, they are “split” into at2_cl12, (0.91) at1_cl13 (AUROC 0.97), at4_cl9 (AUROC 0.94). Theoretically, they should have all a “mutual” hit, but because the ratio is 1:3, it does not happen, and the hit is classified as “non-mutual”. This is a case when the AUROC is still considerably high despite the “non-mutual” correspondence.

To conclude, besides the “mutual” hits, we deem the unidirectional connections beneficial because they help restrict the field of investigation.

3. The spatial distribution pattern of MERFISH clusters is not easy to discriminate due to the mixture of clusters in the figure (Fig 3F, 4A, 4F, 5A, 6A). The authors can present a Voronoi plot showcase of important clusters. Besides, the authors can add the cell density curve of the MERFISH cluster along the medial-lateral and dorsal-ventral axis.

Regarding the first part of the comment: “***The spatial distribution pattern of MERFISH clusters is not easy to discriminate due to the mixture of clusters in the figure***”. We agree that in the mentioned figures, it is not easy to discriminate all the individual clusters, especially the tiny ones, due to their mixture and the high number of detected clusters per anatomical region. However, we still deem that displaying all the clusters at each bregma interval is highly informative for the readers because it gives an overview of the cluster trajectory in x, y, and z dimensions, at least for the main clusters. For readers interested in a higher resolution view of the different clusters, we have provided a freely accessible online interactive spatial viewer at HEAVY.AI [<http://harvard.heavy.ai:6273/>] to explore the clustering at the level of the entire dPnTg and at the subregion level (atlases 1-4).

Regarding the second part of the comment, “***adding the cell density curve of the MERFISH cluster along the medial-lateral and dorsal-ventral axis***”. We calculated the cell density curve, representing the number of cells of each cluster/ total cells across all the rostrocaudal levels (z-axis) of each specific subregion's atlas. This approach helped to visually estimate the cluster frequency distribution along the rostrocaudal axis and facilitated pinpointing the exact level (coronal section) at which the cluster is more frequent. However, it does not supply information on the cluster distribution along the medial-lateral and dorsal-ventral axes. We believe cluster distribution for those levels needs to be calculated and plotted for each section

since the brain anatomy along the rostrocaudal axis changes at each level. Voronoi plots or the spatial viewer can help to visually inspect the medial-lateral and dorsal-ventral cluster density.

4-Previous study has shown the spatial transcriptomics of PBN in mouse (Fu et al., 2022, Cell). The authors should compare their MERFISH data with those as well.

While we agree with the potential value of comparing our study with other studies that have looked at the PB, in this case, for a number of reasons (explained below in detail) and because reviewers and editors have instructed us to decrease the length of the manuscript, we believe it will not be of great benefit to compare our study to that of Fu et al. Below are the details of our reasoning. We evaluated the study on a methodological level, focusing on the study design and assay performance (resolution/ sensitivity/ efficiency), and on a finding level, comparing the results with our MERFISH atlas 2 of the PB.

Briefly, the authors sampled 3 sequential sections at -5.20, -5.35, and -5.50 mm from bregma (Franklin-Paxinos atlas), then used Pixel-Seq, an array-based spatial assay, to profile the whole transcriptome of a total of 31,505 neurons passing the QC that were then used for clustering. This approach harbors several limitations: first, it avoids the very rostral and caudal parts of the PB that, as we and others reported, extends from -4.95 to -5.75 from bregma and not from -5.20 to -5.50; second, it samples a coronal section every 150 microns, whereas our study and the ABA sample every 80 and 100 microns, respectively. Both points are methodological choices that have surely impacted the actual representation of the PB neuron types, potentially causing missing important populations present at the very rostral or caudal part or extending for <150 microns along the z-axis.

In addition, the authors clearly state in the study's limitation section that: *“Due to the timing of developing the stamping method, the OB and PBN data were collected with sequenced gels. Despite the improved feature resolution, DNA array-based spatial transcriptomic assays still face challenges to reliably achieve single-cell resolution. Comparing our clustering results with those on dissociative scRNA-seq of brain tissues, Pixel-seq showed less optimal cell type separation.”* The fact that Pixel-Seq does not achieve single-cell resolution impacted the cluster granularity, i.e. the ability to resolve all the transcriptionally different clusters of the PB. In fact, the Fu et al. study identifies only 18 neuronal subtypes (of which 3 are from neighboring regions), whereas ours identified 36 PB subtypes. The incongruence in terms of the number of clusters identified by the two studies might depend on the abovementioned factors, even though other variables cannot be excluded. Therefore, given the difference in resolution and sensitivity between the two studies, comparing or integrating the datasets would not result in additional information/ insights but might lead to inaccuracies.

5. The author compared the transcriptomic subtypes between mice and humans with snRNA-seq data and found that “33/52 human snRNA-seq clusters corresponded to 28/62 mouse DroNc-seq clusters, and the correspondence was mutual in 23/38 instances (Fig 8H-I; STable 30) (line 601, page 12).” It is not clear how mutual correspondence was calculated. How could the ratio of mutual correspondence be higher than the ratio of unidirectional correspondence?

We invite the reviewer to consult the following two manuscripts (PMID: 29491377; PMID: 34234317) for a comprehensive explanation of the MetaNeighbor package and the functions used to compute “mutual” and “unidirectional” correspondences. In addition, a summarized explanation is in the manuscript's method section and the code used on Zenodo repository.

It is noteworthy to mention that while we were revising our manuscript, the authors from the Siletti et al. paper added further human snRNA-seq data from our ROI to their initial dataset. Therefore, we decided to integrate these new data and redo all downstream analyses. Accordingly, in our revised manuscript, we have updated the relative results section (lines 425-467), Supp. Figures (12-14), Supp. Tables (27-33) and MetaNeighbor results. In the latest MetaNeighbor analysis using our updated snRNA-seq datasets, we found that 50/67 human snRNA-seq neuronal clusters corresponded to 52/127 mouse snRNA-seq neuronal clusters, and the correspondence was “mutual” in 23/64 instances. Specifically, when we refer to 50/67 human clusters corresponding to 52/127 mouse clusters, we mean that 50 and 52 clusters have a match in the corresponding dataset that can be “mutual” or “non-mutual”. Overall, the connections that passed the AUROC threshold are in total 64, of which 23 are “mutual” and 41 are “non-mutual” (i.e., unidirectional). The clusters excluded made a match, but the AUROC value did not pass the threshold set. Ideally, if there would be perfect conservation between the two species, the same precision in the dissections with the inclusion of identical brain structures, and the same number of clusters in both datasets, then all clusters should be mapped as “mutual” correspondences, that is, 1:1 and have an AUROC score ~1. However, in real interspecies comparison, the “mutual” matches are often less than “non-mutual” for the abovementioned reasons and other minor variables. Finally, we hypothesized possible reasons behind the fact that ~80% (50/67) of human clusters versus only ~40% (52/127) of mouse clusters made a match. We speculated that: **i- the imbalance in cluster composition between the two datasets may originate from the difference in precision between human and mouse dissections.** The difference could stem from the fact that dissecting the human brain is a relatively more precise process than doing microdissections in the mouse brain, simply because the human brain is 3×10^3 times bigger. Therefore, the dissection precision between humans and mice could explain part of this imbalance. In addition, dissections done by different groups at different times with different goals almost certainly did not include exactly the same structures. Clusters not making any link could be part of contaminant/ neighbor regions present in the mouse dissection but absent in the human one. **ii- some of the mouse clusters are not conserved, or their transcriptional profile diverges significantly from the respective cluster in humans.** Part of the clusters that do not form any link in mice could not be conserved between species, or their transcriptional profile can diverge significantly. **iii- our conservative AUROC threshold could have excluded “less-conserved” correspondences.** This means that lowering the threshold will allow the detection of part of these more transcriptionally divergent clusters but could also introduce false positives in the analysis.

6. The authors mentioned, “All neurons in this group expressed Slc32a1 (Vgat).” (line 142, page 3). However, the

dot plot (Fig 1L) didn't support this conclusion because the ratio of *Slc32a1*+ neurons in these named "inhibitory neurons groups" is much smaller than 1.

The reviewer's observation is correct. This was maybe due to a combination of factors such as the high drop-out particularly affecting *Slc32a1* (*Vgat*) and/or ambient RNA contamination more evident in lower-quality nuclei (nuclei with lower UMIs per gene). Raising the low threshold to exclude those nuclei and including a dataset from the latest 10X technology has improved the rate of gene detection, especially for those genes that are low-expressed / or with low turnover. However, we also acknowledge that some of these clusters could be actual *Vgat*+/*Vglut2*+ ("hybrid") neurons, as shown in Fig-4 for clusters 12, 14, 33, and as documented in the literature. Accordingly, in the revised manuscript, we have rearranged the section related to the snRNA-seq analysis of the mouse dPnTg dataset (lines 91--127) and relative figures (Fig1 manuscript) and Supp. Tables (n2-3).

Reviewer #2 (Remarks to the Author):

In this manuscript, the authors make use of two powerful technologies, MERFISH, and single-nucleus (sn) transcriptional profiling to obtain a very powerful and highly resolved map of the dorsal pontine tegmentum (dPnTg) of the murine brain. They subsequently use established methods and approaches to cluster and subcluster the different cell types within the dPnTg and performed pairwise comparisons for cluster replicability across both technologies and with data from human snRNA-seq samples. Of note, to visualize and make this large dataset more accessible the authors also invested time in creating an interactive dashboard which contains selection tools, zoom functions and several informative plots that display metadata. Overall, the open-access repository of all the raw data and of the visualized pre-processed data serves as an invaluable resource, empowering researchers in this field and has great potential to facilitate rapid unraveling of novel functions and biological processes within this intricate brain tissue.

Overall, I applaud the authors for making such a detailed and unprecedented dataset and for their diligence in disentangling all the different cell types within the dPnTg. I do have a few major and minor comments that I hope can further improve the readability and strength of the manuscript.

Major comments:

My main issue concerns the readability of the paper. Each figure contains a lot of panels and information, and it's extremely difficult to read the details. I believe it would be beneficial to reduce the number of panels in the main figures and move supporting figures to supplemental data. Similarly, it would help if the authors could provide more schematics for some of the figures that would depict their experimental or data analysis strategy. For example, in Figure 2, they display a simplistic MERFISH pipeline, but it would help more if they could explain how many mice and serial sections (80-90 μm intervals) they took from each. Is this, in fact, a (serial) 3D dataset from the same mouse or an aggregation of different mice?

We fully agree with the reviewer's comments. We made the following modifications to the manuscript to improve its readability according to the author's guidelines of Nature Communication and the reviewer's comments: i- we have shortened the paper from >8,000 to <5,500 words; ii- we have reduced the complexity and number of the main figures/tables by adding relevant panels in the supplementary section; iii- we have improved the schematics of

MERFISH and snRNA-seq assays to give a more detailed overview of the experimental workflow. This has improved the quality and the form of the manuscript.

Regarding the specific question raised. We applied MERFISH to a total of 7 C57BL/6J mice. From 4 of them (2 males and 2 females), we cut 10 sequential, 10-micron-thick sections at an interval of 80-90 microns one from each other, spanning a region from approximately -4.70 to -5.80 bregma level in the Franklin-Paxinos atlas. We profiled 46 coronal sections, of which 40 are sequential sections from 4 mice (10 sequential sections from each mouse), and 6 are spare sections from other 3 mice. They are from pilot experiments, and since the quality of the MERFISH assay and the brain region sampled were optimal, they were included in the final dataset. We have added this relevant information to the method section and the schematic in Fig. 2 of the manuscript.

Minor comments:

1. Most of the analysis strategy focuses on cell type identification (and cross-data comparison), and there is a missed opportunity to explore the presence of spatial patterns (genes, niches, regions, domains, ...). Can the serial sections be considered a 3D dataset and be used to identify brain structures and expression in 3 dimensions? Besides visualizations of cell type distribution changes within the sections, there hasn't been much done. Is this the consequence of issues with registering the serial images?

The point raised by the reviewer is a legitimate observation. Since the sections are sequential and originate from the same mouse, they can be used to identify brain structures and gene expression patterns in 3D. There is no issue registering the images. During the review process, one of our collaborators, Dr. Anderman, registered a series of 10 MERFISH sequential sections onto the Allen Brain Atlas framework. We are also working with HEAVY.AI to build a 3D atlas of this region (there are no prototypes available to the scientific community, as far as we are aware), but we are still far from its completion, and we could not include it in this manuscript.

2. The GPU-accelerated visualization dashboard is potentially a powerful tool for researchers to display similar datasets in the future; however, it is unclear whether this dashboard is specifically tailored for this dataset or could also be used with external datasets.

This dashboard has been designed specifically for this dataset. However, HEAVY.AI proprietary technology could surely be used to display other spatial transcriptomics or single-cell datasets far larger in the future.

REVIEWERS' COMMENTS

Reviewer #1 (Remarks to the Author):

The authors have addressed all the concerns. No more comments.

Reviewer #2 (Remarks to the Author):

The authors answered all my questions and significantly improved the manuscript. I have no further comments.

REVIEWERS COMMENTS

Overall, we thank the reviewers and editor for their very constructive comments. This led to a number of changes and improvements, which we believe have greatly enhanced the quality of our study. A bullet list of all our revision's major changes/ improvements is summarized immediately below. After that are our detailed responses to each of the reviewers' comments.

Summary of major changes/ improvements to the manuscript.

1. Integration of our DroNc-seq data with Allen Brain Atlas snRNA-seq data (10X.v3) from the same region of interest, the mouse dPnTg, has significantly improved the statistical power, snRNA-seq metrics, clustering granularity, and correlation with the MERFISH dataset.
2. Transcriptional correspondence of neuronal clusters from MERFISH atlases 1-4 with snRNA-seq neuronal clusters of the mouse dPnTg has been assessed by MetaNeighbor analysis. Results have been replicated by canonical correlation analysis (CCA) computed at the single-cell level.
3. Cell clustering and related analyses on the human dPnTg snRNA-seq dataset have been re-performed after integrating additional data made available in the Siletti et al. manuscript. MetaNeighbor-inferred correspondences between mouse and human snRNA-seq clusters of the dPnTg have also been recomputed.
4. The manuscript has been drastically shortened from over 8,000 to <5,500 words and modified according to the Nature Communication guidelines and the reviewer's comments.
5. Figures and tables have been simplified by reducing the number of panels while keeping the most relevant insights. Non-central findings and all technical analyses have been moved into the supplementary methods.
6. The figure schematics for the snRNAseq and MERFISH studies have been redesigned to give a quick overview of the experimental flowchart. Implementing points n. 4-6 has significantly improved the manuscript's readability.
7. BROAD single-cell and HEAVY.AI spatial dashboards plotting mouse snRNA-seq and MERFISH data have been updated according to the new analysis to allow the reviewers to access the data easily.
8. Finally, we have made available raw/ preprocessed data and the code used for the analyses, and we have tested and confirmed the reproducibility of the results.

Reviewer #1 (Remarks to the Author):

In this manuscript, the authors performed single-nucleus RNA-seq (snRNA-seq) and generated a transcriptomic atlas of the "dorsal pontine tegmentum" (dPnTg) of the mouse. Further, the authors employed MERFISH to map the distribution of 315 informative genes hints from snRNA-seq to map the subtypes spatially. They compared the neuronal subtypes of dPnTg between mice and humans and found that many subtypes were transcriptionally similar. Finally, the authors developed a freely-accessible website to provide access to this spatially-resolved transcriptional dataset. This study represents a valuable resource for further study of the function of these nuclei. This reviewer has several concerns.

1. The authors profiled 447,833 nuclei, but only 149,159 nuclei finally passed the quality control steps (line 109, page 3). Why were two-thirds of nuclei unqualified? Does this mean the quality of snRNA-seq is low?

The reviewer raised an important point. We provide our answer to this question in the following points below.

1. Generally, single-nuclei RNA-seq detects fewer genes per nucleus than single-cell RNA-seq, as only the nuclear fraction of the transcriptome is sequenced. In fact, the number of transcripts in the nucleus represents ~30% of the total mRNA fraction, depending on the cell type (PMID: 30586455). Consequently, it is more likely that more nuclei than whole cells are discarded in QC steps that involve filtering out nuclei with a low gene detection rate. We also highlight that the number of detected genes per nucleus set as a cutoff to exclude potential empty droplets/ low-quality nuclei from downstream analyses was in line with the commonly used cutoff values.

Notably, despite 2/3 of the nuclei being discarded, the number of nuclei used (N=149,159) for downstream analyses was still higher than most of the other published single-cell RNAseq atlases (Fig-1).

2. In addition, profiling nuclei transcriptomes offers more advantages than whole cells in neuroscience research. For instance, it avoids alterations in transcription, such as the expression of early-immediate genes (IEGs), that could distort the actual cellular transcriptional profile; it also avoids the possibility of some of the larger cells being excluded during the filtering process and/ or failing to survive the dissociation steps, which prevents bias in cell type representation. Nowadays, most sequencing studies in neuroscience are performed on nuclei.

3. Our DroNc-seq assay was performed as the first step of this study using an in-house protocol. Noteworthy, we used the same setup adopted in the Aviv Regev Lab (PMID: 28846088). Our QC'ed dataset shows a gene detection rate highly consistent with the abovementioned paper (mean~1,300 genes/nucleus). While we are aware that in these past 5 years, the single-nuclei protocols have greatly improved, leading to better assay sensitivity and efficiency, we would also highlight that our QC'ed dataset (N=149,159 nuclei) provided sufficient granularity and statistical power to transcriptionally characterize, in detail, our region of interest. We also reasoned that increasing the stringency of our cutoff could be beneficial despite further reducing the number of nuclei available for analyses. To this end, we run cell Bender (PMID: 37550580) on the DroNc-seq data to estimate a threshold of a number of genes per nucleus capable of including only true positive nuclei. Cell Bender suggested discarding nuclei with <400 detected genes, which resulted in a dataset of ~95,000 nuclei. We used this dataset as input to an integration procedure outlined in point 4 below.

4. Finally, for this revision, we put effort into increasing the sample size and further improving the snRNA-seq metrics of the study by integrating our DroNc-seq (~95,000 dPnTg nuclei) data with a more recent snRNA-seq (10X_v3) (~125,000 dPnTg nuclei) dataset from the ongoing Allen Brain Atlas (ABA) study sampling the same regions of the pons [doi: <https://doi.org/10.1101/2023.03.06.531121>; doi: <https://doi.org/10.1101/2023.03.06.531348>]. DroNc-seq data integration with the ABA snRNA-seq dataset and the use of a more stringent cutoff for downstream analyses, as described in point 3, improved the sample size and statistical power of the study, the snRNA-seq metrics, clustering granularity, the ability to detect low-expressed genes/ genes with low turnover and the cluster-to-cluster correspondence with MERFISH dataset (MetaNeighbor analysis). More importantly, in our revised dataset, the number of nuclei used for downstream analyses is higher than those from published single-cell RNAseq atlases that investigated brain regions of similar size, and the median number of genes per cell aligns with recent field works (Fig-1).
5. Accordingly, in the revised manuscript, we have rearranged the result section related to the mouse dPnTg cell type identification using snRNA-seq (lines 92- 128) and updated all relative panels in Fig 1 (except panel 1b), Supp. Fig. 1-2 and Supp. Tables 1-3. Likewise, results from MetaNeighbor analysis performed between snRNA-seq neuronal clusters of the mouse dPnTg versus the cluster annotation of the 4 MERFISH atlases (Fig. 8a-b; Supp. Table 26) and versus the cluster annotation of the human dPnTg (Supp Fig 12-14; Supp. Tables 30-31) have been updated as well and moved to line 411-- 424 and line 441– 468, respectively.

2. To reveal the correspondence between MERFISH and DroNc-seq clusters of the dPnTg, the authors applied an unsupervised replication framework and depicted the clusters correspondence between MERFISH and DroNc-seq datasets (Fig 2M). However, this cluster-to-cluster correspondence lost the single-cell solution. Is it possible to examine the correspondence with single-cell resolution? Besides, the authors mentioned, “the correspondence between MERFISH and DroNc-seq clusters was “mutual” in 38/84 instances (line 211, page 5)”. The ratio of “mutual” correspondence is less than 0.5. Can the author speculate on the possible reason?

The reviewer raised two constructive points that we have addressed below as follows:

1. **“loss of single-cell resolution when applying**

MetaNeighbor to assess cluster-to-cluster correspondence between MERFISH and snRNA-seq datasets". MetaNeighbor is optimized to give back robust statistics, which operates better at the cluster level. Theoretically, it would be possible to run it on individual cells, but there would be no way of scoring the goodness of the integration because the statistic, in this case, relies on how well the cells are grouped. In addition, because MetaNeighbor does not use any threshold, a cell would not be automatically placed into one category instead of another. We deem that a correspondence computed at the cluster level is correct and robust enough to impute transcriptional information between the two datasets. However, for visualization purposes, such as in our HEAVY.AI dashboard, where data are displayed at the single-cell level, we performed a canonical correlation analysis (CCA) at the single-cell level. We then aggregated the single-cell results at the cluster level to calculate the cluster-to-cluster correspondence and compare the MetaNeighbor output with the CCA output (see methods for a detailed explanation of thresholds). The two methods agreed in 73/105 (70%) instances. (Fig-2). Setting a more stringent threshold would significantly increase the agreement between the two methods, even though the number of investigated matches would drop.

2. **"The ratio of "mutual" correspondence is <0.5"**

This point requires a detailed explanation. In MetaNeighbor, we set "symmetric_output= FALSE", meaning each cluster is tested against only two clusters in the target dataset (closest and second-closest match). This representation helps to rapidly identify a cluster's closest hits as well as its closest outgroup. The nonsymmetric view makes it clear when best hits are not reciprocal. The lack of reciprocity in voting is an important tool for detecting imbalances in dataset composition. In our new MetaNeighbor analysis

(line 410-423; Fig 8 manuscript) performed between the neuronal clusters of the 4 MERFISH atlases and snRNAseq (10X + DroNc-seq) neuronal clusters, we have a "mutual" match in 50/122 instances (40%). This result might depend on several factors: **i- the AUROC threshold set.** In Fig-3, we show the distribution of all the matches with an AUROC > 0.85 that are classified as "mutual" or "non-mutual" hits. "Non-mutual" hits have a lower median and show a bimodal distribution. Therefore, raising the threshold to include matches >0.9 (instead of >0.85) and/or "mutual" hits will drastically improve the % of "mutual" hits over the total hits, bringing it to >55%. However, this will cause the exclusion of some biologically relevant, "non-mutual" correspondences; **ii- differences in the region of interest (ROI) sampled in snRNA-seq (10X.v3 + DroNc-seq) versus MERFISH atlases 1-4.** Specifically, stereotactic dissections of the dPnTg done by this study and the ABA effort included unavoidably marginal regions external to our ROI that could not be excluded *a priori* since single cells do not hold any spatial information. On the contrary, MERFISH clusters from atlases 1-4 mapped only to our ROI, the dPnTg, since it was possible to filter out clusters from neighbor regions, given the technology holds spatial information. Clusters that, for this reason, will not have their corresponding match in the comparing dataset will map onto the closest transcriptional cluster, but the match's AUROC score will be lower (since they are not transcriptionally identical), and the match will be "non-mutual". Most of these matches will not pass the AUROC threshold set, though.

iii- the difference in cluster granularity between the two approaches leads to fewer "mutual" matches. The difference in the number of variable features used to cluster the two datasets (315 vs. 3000) led to snRNA-seq resolving more (>30%) neuron types

compared to MERFISH when we clustered the neurons by dividing them into “excitatory” and “inhibitory” groups (Figs 1-2 manuscript). To partially mitigate the difference in cluster granularity between the two technologies, we compared the snRNA-seq neuronal clustering with the clusters from the 4 non-overlapping MERFISH subregion atlases that achieved a better granularity because clustering was done on smaller regions than the whole PnTg (lines 410—423; Fig 8; STable 26 manuscript). This led to identifying more “mutual” matches and gave a more resolved representation of neuronal types' transcriptional and spatial organization in the dPnTg. Regarding “non-mutual” matches, either a MERFISH cluster corresponded to more than one snRNA-seq cluster or viceversa. In both scenarios, one technology could resolve one or more neuronal types better than the other. In the first scenario, snRNA-seq led to better granularity than MERFISH; a possibility would be that more snRNAseq clusters matching with only one MERFISH cluster represent subclusters of the main MERFISH population that could not be resolved into subpopulations for the absence of one or more marker genes. The opposite scenario happens because clustering smaller regions, as we did for the 4 MERFISH atlases, also leads to better cluster granularity than clustering the whole dPnTg. **iv) transcriptionally identical neuron types in the 4 MERFISH atlases lead to “non-mutual” matches with snRNAseq.** A clear example is the Sst neurons, which in the snRNAseq dataset cluster all together (cluster_ex_33), whereas in the MERFISH dataset, composed of 4 atlases clustered independently, they are “split” into at2_cl12, (0.91) at1_cl13 (AUROC 0.97), at4_cl9 (AUROC 0.94). Theoretically, they should have all a “mutual” hit, but because the ratio is 1:3, it does not happen, and the hit is classified as “non-mutual”. This is a case when the AUROC is still considerably high despite the “non-mutual” correspondence.

To conclude, besides the “mutual” hits, we deem the unidirectional connections beneficial because they help restrict the field of investigation.

3. The spatial distribution pattern of MERFISH clusters is not easy to discriminate due to the mixture of clusters in the figure (Fig 3F, 4A, 4F, 5A, 6A). The authors can present a Voronoi plot showcase of important clusters. Besides, the authors can add the cell density curve of the MERFISH cluster along the medial-lateral and dorsal-ventral axis.

Regarding the first part of the comment: “***The spatial distribution pattern of MERFISH clusters is not easy to discriminate due to the mixture of clusters in the figure***”. We agree that in the mentioned figures, it is not easy to discriminate all the individual clusters, especially the tiny ones, due to their mixture and the high number of detected clusters per anatomical region. However, we still deem that displaying all the clusters at each bregma interval is highly informative for the readers because it gives an overview of the cluster trajectory in x, y, and z dimensions, at least for the main clusters. For readers interested in a higher resolution view of the different clusters, we have provided a freely accessible online interactive spatial viewer at HEAVY.AI [<http://harvard.heavy.ai:6273/>] to explore the clustering at the level of the entire dPnTg and at the subregion level (atlases 1-4).

Regarding the second part of the comment, “***adding the cell density curve of the MERFISH cluster along the medial-lateral and dorsal-ventral axis***”. We calculated the cell density curve, representing the number of cells of each cluster/ total cells across all the rostrocaudal levels (z-axis) of each specific subregion's atlas. This approach helped to visually estimate the cluster frequency distribution along the rostrocaudal axis and facilitated pinpointing the exact level (coronal section) at which the cluster is more frequent. However, it does not supply information on the cluster distribution along the medial-lateral and dorsal-ventral axes. We believe cluster distribution for those levels needs to be calculated and plotted for each section

since the brain anatomy along the rostrocaudal axis changes at each level. Voronoi plots or the spatial viewer can help to visually inspect the medial-lateral and dorsal-ventral cluster density.

4-Previous study has shown the spatial transcriptomics of PBN in mouse (Fu et al., 2022, Cell). The authors should compare their MERFISH data with those as well.

While we agree with the potential value of comparing our study with other studies that have looked at the PB, in this case, for a number of reasons (explained below in detail) and because reviewers and editors have instructed us to decrease the length of the manuscript, we believe it will not be of great benefit to compare our study to that of Fu et al. Below are the details of our reasoning. We evaluated the study on a methodological level, focusing on the study design and assay performance (resolution/ sensitivity/ efficiency), and on a finding level, comparing the results with our MERFISH atlas 2 of the PB.

Briefly, the authors sampled 3 sequential sections at -5.20, -5.35, and -5.50 mm from bregma (Franklin-Paxinos atlas), then used Pixel-Seq, an array-based spatial assay, to profile the whole transcriptome of a total of 31,505 neurons passing the QC that were then used for clustering. This approach harbors several limitations: first, it avoids the very rostral and caudal parts of the PB that, as we and others reported, extends from -4.95 to -5.75 from bregma and not from -5.20 to -5.50; second, it samples a coronal section every 150 microns, whereas our study and the ABA sample every 80 and 100 microns, respectively. Both points are methodological choices that have surely impacted the actual representation of the PB neuron types, potentially causing missing important populations present at the very rostral or caudal part or extending for <150 microns along the z-axis.

In addition, the authors clearly state in the study's limitation section that: "*Due to the timing of developing the stamping method, the OB and PBN data were collected with sequenced gels. Despite the improved feature resolution, DNA array-based spatial transcriptomic assays still face challenges to reliably achieve single-cell resolution. Comparing our clustering results with those on dissociative scRNA-seq of brain tissues, Pixel-seq showed less optimal cell type separation.*" The fact that Pixel-Seq does not achieve single-cell resolution impacted the cluster granularity, i.e. the ability to resolve all the transcriptionally different clusters of the PB. In fact, the Fu et al. study identifies only 18 neuronal subtypes (of which 3 are from neighboring regions), whereas ours identified 36 PB subtypes. The incongruence in terms of the number of clusters identified by the two studies might depend on the abovementioned factors, even though other variables cannot be excluded. Therefore, given the difference in resolution and sensitivity between the two studies, comparing or integrating the datasets would not result in additional information/ insights but might lead to inaccuracies.

5. The author compared the transcriptomic subtypes between mice and humans with snRNA-seq data and found that "33/52 human snRNA-seq clusters corresponded to 28/62 mouse DroNc-seq clusters, and the correspondence was mutual in 23/38 instances (Fig 8H-I; STable 30) (line 601, page 12)." It is not clear how mutual correspondence was calculated. How could the ratio of mutual correspondence be higher than the ratio of unidirectional correspondence?

We invite the reviewer to consult the following two manuscripts (PMID: 29491377; PMID: 34234317) for a comprehensive explanation of the MetaNeighbor package and the functions used to compute "mutual" and "unidirectional" correspondences. In addition, a summarized explanation is in the manuscript's method section and the code used on Zenodo repository.

It is noteworthy to mention that while we were revising our manuscript, the authors from the Siletti et al. paper added further human snRNA-seq data from our ROI to their initial dataset. Therefore, we decided to integrate these new data and redo all downstream analyses. Accordingly, in our revised manuscript, we have updated the relative results section (lines 425-467), Supp. Figures (12-14), Supp. Tables (27-33) and MetaNeighbor results. In the latest MetaNeighbor analysis using our updated snRNA-seq datasets, we found that 50/67 human snRNA-seq neuronal clusters corresponded to 52/127 mouse snRNA-seq neuronal clusters, and the correspondence was “mutual” in 23/64 instances. Specifically, when we refer to 50/67 human clusters corresponding to 52/127 mouse clusters, we mean that 50 and 52 clusters have a match in the corresponding dataset that can be “mutual” or “non-mutual”. Overall, the connections that passed the AUROC threshold are in total 64, of which 23 are “mutual” and 41 are “non-mutual” (i.e., unidirectional). The clusters excluded made a match, but the AUROC value did not pass the threshold set. Ideally, if there would be perfect conservation between the two species, the same precision in the dissections with the inclusion of identical brain structures, and the same number of clusters in both datasets, then all clusters should be mapped as “mutual” correspondences, that is, 1:1 and have an AUROC score ~1. However, in real interspecies comparison, the “mutual” matches are often less than “non-mutual” for the abovementioned reasons and other minor variables. Finally, we hypothesized possible reasons behind the fact that ~80% (50/67) of human clusters versus only ~40% (52/127) of mouse clusters made a match. We speculated that: **i- the imbalance in cluster composition between the two datasets may originate from the difference in precision between human and mouse dissections.** The difference could stem from the fact that dissecting the human brain is a relatively more precise process than doing microdissections in the mouse brain, simply because the human brain is 3×10^3 times bigger. Therefore, the dissection precision between humans and mice could explain part of this imbalance. In addition, dissections done by different groups at different times with different goals almost certainly did not include exactly the same structures. Clusters not making any link could be part of contaminant/ neighbor regions present in the mouse dissection but absent in the human one. **ii- some of the mouse clusters are not conserved, or their transcriptional profile diverges significantly from the respective cluster in humans.** Part of the clusters that do not form any link in mice could not be conserved between species, or their transcriptional profile can diverge significantly. **iii- our conservative AUROC threshold could have excluded “less-conserved” correspondences.** This means that lowering the threshold will allow the detection of part of these more transcriptionally divergent clusters but could also introduce false positives in the analysis.

6. The authors mentioned, “All neurons in this group expressed Slc32a1 (Vgat).” (line 142, page 3). However, the

dot plot (Fig 1L) didn't support this conclusion because the ratio of *Slc32a1*+ neurons in these named "inhibitory neurons groups" is much smaller than 1.

The reviewer's observation is correct. This was maybe due to a combination of factors such as the high drop-out particularly affecting *Slc32a1* (*Vgat*) and/or ambient RNA contamination more evident in lower-quality nuclei (nuclei with lower UMIs per gene). Raising the low threshold to exclude those nuclei and including a dataset from the latest 10X technology has improved the rate of gene detection, especially for those genes that are low-expressed / or with low turnover. However, we also acknowledge that some of these clusters could be actual *Vgat*+/*Vglut2*+ ("hybrid") neurons, as shown in Fig-4 for clusters 12, 14, 33, and as documented in the literature. Accordingly, in the revised manuscript, we have rearranged the section related to the snRNA-seq analysis of the mouse dPnTg dataset (lines 91--127) and relative figures (Fig1 manuscript) and Supp. Tables (n2-3).

Reviewer #2 (Remarks to the Author):

In this manuscript, the authors make use of two powerful technologies, MERFISH, and single-nucleus (sn) transcriptional profiling to obtain a very powerful and highly resolved map of the dorsal pontine tegmentum (dPnTg) of the murine brain. They subsequently use established methods and approaches to cluster and subcluster the different cell types within the dPnTg and performed pairwise comparisons for cluster replicability across both technologies and with data from human snRNA-seq samples. Of note, to visualize and make this large dataset more accessible the authors also invested time in creating an interactive dashboard which contains selection tools, zoom functions and several informative plots that display metadata. Overall, the open-access repository of all the raw data and of the visualized pre-processed data serves as an invaluable resource, empowering researchers in this field and has great potential to facilitate rapid unraveling of novel functions and biological processes within this intricate brain tissue.

Overall, I applaud the authors for making such a detailed and unprecedented dataset and for their diligence in disentangling all the different cell types within the dPnTg. I do have a few major and minor comments that I hope can further improve the readability and strength of the manuscript.

Major comments:

My main issue concerns the readability of the paper. Each figure contains a lot of panels and information, and it's extremely difficult to read the details. I believe it would be beneficial to reduce the number of panels in the main figures and move supporting figures to supplemental data. Similarly, it would help if the authors could provide more schematics for some of the figures that would depict their experimental or data analysis strategy. For example, in Figure 2, they display a simplistic MERFISH pipeline, but it would help more if they could explain how many mice and serial sections (80-90 μm intervals) they took from each. Is this, in fact, a (serial) 3D dataset from the same mouse or an aggregation of different mice?

We fully agree with the reviewer's comments. We made the following modifications to the manuscript to improve its readability according to the author's guidelines of Nature Communication and the reviewer's comments: i- we have shortened the paper from >8,000 to <5,500 words; ii- we have reduced the complexity and number of the main figures/tables by adding relevant panels in the supplementary section; iii- we have improved the schematics of

MERFISH and snRNA-seq assays to give a more detailed overview of the experimental workflow. This has improved the quality and the form of the manuscript.

Regarding the specific question raised. We applied MERFISH to a total of 7 C57BL/6J mice. From 4 of them (2 males and 2 females), we cut 10 sequential, 10-micron-thick sections at an interval of 80-90 microns one from each other, spanning a region from approximately -4.70 to -5.80 bregma level in the Franklin-Paxinos atlas. We profiled 46 coronal sections, of which 40 are sequential sections from 4 mice (10 sequential sections from each mouse), and 6 are spare sections from other 3 mice. They are from pilot experiments, and since the quality of the MERFISH assay and the brain region sampled were optimal, they were included in the final dataset. We have added this relevant information to the method section and the schematic in Fig. 2 of the manuscript.

Minor comments:

1. Most of the analysis strategy focuses on cell type identification (and cross-data comparison), and there is a missed opportunity to explore the presence of spatial patterns (genes, niches, regions, domains, ...). Can the serial sections be considered a 3D dataset and be used to identify brain structures and expression in 3 dimensions? Besides visualizations of cell type distribution changes within the sections, there hasn't been much done. Is this the consequence of issues with registering the serial images?

The point raised by the reviewer is a legitimate observation. Since the sections are sequential and originate from the same mouse, they can be used to identify brain structures and gene expression patterns in 3D. There is no issue registering the images. During the review process, one of our collaborators, Dr. Anderman, registered a series of 10 MERFISH sequential sections onto the Allen Brain Atlas framework. We are also working with HEAVY.AI to build a 3D atlas of this region (there are no prototypes available to the scientific community, as far as we are aware), but we are still far from its completion, and we could not include it in this manuscript.

2. The GPU-accelerated visualization dashboard is potentially a powerful tool for researchers to display similar datasets in the future; however, it is unclear whether this dashboard is specifically tailored for this dataset or could also be used with external datasets.

This dashboard has been designed specifically for this dataset. However, HEAVY.AI proprietary technology could surely be used to display other spatial transcriptomics or single-cell datasets far larger in the future.